# Cytotoxic T-cells mediate exercise-induced reductions in tumor growth

Helene Rundqvist[1,2], Pedro Veliça[1], Laura Barbieri[1,3], Paulo A Gameiro[4], David Bargiela[1,5], Milos Gojkovic[1], Sara Mijwel[6], Stefan Markus Reitzner[6], David Wulliman[1], Emil Ahlstedt[2], Jernej Ule[4], Arne Östman[7], Randall S Johnson[1,5]*

[1]Department of Cell and Molecular Biology, Karolinska Institutet, Stockholm, Sweden; [2]Department of Laboratory Medicine, Karolinska Institutet, Stockholm, Sweden; [3]Department of Surgery, Oncology, and Gastroenterology, University of Padova, Padua, Italy; [4]The Francis Crick Institute, London, United Kingdom; [5]Department of Physiology, Development, and Neuroscience, University of Cambridge, Cambridge, United Kingdom; [6]Department of Physiology and Pharmacology, Karolinska Institutet, Stockholm, Sweden; [7]Department of Oncology-Pathology, Karolinska Institutet, Stockholm, Sweden

**Abstract** Exercise has a wide range of systemic effects. In animal models, repeated exertion reduces malignant tumor progression, and clinically, exercise can improve outcome for cancer patients. The etiology of the effects of exercise on tumor progression are unclear, as are the cellular actors involved. We show here that in mice, exercise-induced reduction in tumor growth is dependent on CD8+ T cells, and that metabolites produced in skeletal muscle and excreted into plasma at high levels during exertion in both mice and humans enhance the effector profile of CD8 + T-cells. We found that activated murine CD8+ T cells alter their central carbon metabolism in response to exertion in vivo, and that immune cells from trained mice are more potent antitumor effector cells when transferred into tumor-bearing untrained animals. These data demonstrate that CD8+ T cells are metabolically altered by exercise in a manner that acts to improve their antitumoral efficacy.

*For correspondence:
rsj33@cam.ac.uk

**Competing interests:** The authors declare that no competing interests exist.

## Introduction

In humans, exercising cohorts have lower rates of cancer incidence (*Moore et al., 2016*) and better outcomes across a range of cancer diagnoses (*Cormie et al., 2017*; *Friedenreich et al., 2016*), proportionate to the degree and intensity of exercise. The mechanisms underlying these observations have remained elusive, although recent work has indicated a relationship between immune response and exercise-induced changes in malignant progression (*Pedersen et al., 2016*; *Koelwyn et al., 2017*).

The metabolic demands of strenuous physical exertion generally induce significant changes in nutrient utilization, principally via central carbon metabolism (*Brooks, 1998*). These exercise-induced alterations in metabolism change the ratios of energy substrates utilized, and can shift intramuscular metabolite profiles. These shifts are reflected in systemic metabolite availability, which in turn modifies energy production throughout the body (*Henderson et al., 2004*; *Lezi et al., 2013*; *El Hayek et al., 2019*).

It is clear that cytotoxic T cells play a crucial role in controlling tumor growth. By recognizing mutation-derived neoantigens, T cells can identify and eliminate malignant cells in a process known as immunosurveillance (*Dunn et al., 2002*; *Dunn et al., 2004*). Escape from immune control is a

**eLife digest** Exercise affects almost all tissues in the body, and scientists have found that being physically active can reduce the risk of several types of cancer as well as improving outcomes for cancer patients. However, it is still unknown how exercise exerts its protective effects. One of the hallmarks of cancer is the ability of cancer cells to evade detection by the immune system, which can in some cases stop the body from eliminating tumor cells.

Rundqvist et al. used mice to investigate how exercise helps the immune system act against tumor progression. They found that when mice exercised, tumor growth was reduced, and this decrease in growth depended on the levels of a specific type of immune cell, the CD8+ T cell, circulating in the blood. Additionally, Rundqvist et al. found that CD8+ T cells were made more effective by molecules that muscles released into the blood during exercise. Isolating immune cells after intense exercise showed that these super-effective CD8+ T cells alter how they use molecules for energy production after exertion. Next, immune cells from mice that had exercised frequently were transferred into mice that had not exercised, where they were more effective against tumor cells than the immune cells from untrained mice.

These results demonstrate that CD8+ T cells are altered by exercise to improve their effectiveness against tumors. The ability of T cells to identify and eliminate cancer cells is essential to avoid tumor growth, and is one of the foundations of current immune therapy treatments. Exercise could improve the outcome of these treatments by increasing the activation of the immune system, making tumor-fighting cells more effective.

critical step toward progressive malignant growth in many cancers, and tumors achieve this in a number of ways, amongst them the dampening of antitumor T cell responses (*Dunn et al., 2002*; *Dunn et al., 2004*; *Beatty and Gladney, 2015*; *Zappasodi et al., 2018*).

The activity of immune cells is tightly linked with their metabolism (*O'Neill et al., 2016*; *Pearce and Pearce, 2013*). Many aspects of immune cell energetics are likely sensitive to the metabolic changes induced by exercise (*Henderson et al., 2004*). Exercise is known to affect immune cell function, and an altered immune response has been suggested as a mechanism underlying effects of exercise on cancer risk and progression (*Christensen et al., 2018*).

In this study, we investigate the association between exercise, tumor growth, and CD8+ T-cell function. To address this, we undertook studies of exercise-induced changes in tumor progression, and asked what metabolites are released in response to exercise; as well as whether metabolites produced by exercise can alter cytotoxic T-cell function. We found that exercise itself can modify cytotoxic T-cell metabolism, and that exercise-induced effects on tumor growth are dependent on cytotoxic T-cell activity.

## Results

### Depletion of CD8+ T cells abolishes the anticancer effects of exercise training

To address the role of immunity on the effects of exercise in neoplasia, we first assessed how repeated voluntary exertion influenced tumor progression in mice, using a genetic model of mammary cancer induced by the MMTV-PyMT transgene on the FVB inbred strain background (*Figure 1—figure supplement 1A*). FVB inbred mice are enthusiastic runners relative to most other inbred strains (*Avila et al., 2017*), and the MMTV-PyMT model in many regard mimics the gradual progression of human breast cancer (*Lin et al., 2003*). PyMT+ mice ran on average 6 km/day (*Figure 1—figure supplement 1B*). Contrary to what was previously shown (*Goh et al., 2013*), the running mice in our experiments showed no statistically significant differences in tumor growth between the groups (*Figure 1—figure supplement 1C- G*). However, the infiltration of Granzyme B (GZMB)-positive cells was significantly higher in primary tumors of running mice, even though voluntary running had no effect on the frequency of CD3, F4/80, PCNA, or podocalyxin-expressing cells (*Figure 1—figure supplement 1H*). This indicates that exercise in this tumor model modulates the infiltration of cytotoxic lymphocytes.

Given that infiltration of cytotoxic T cells is linked to a favorable prognosis in many human neoplasms and in some animal tumor models (*Savas et al., 2016*), we proceeded to explore the role of lymphoid cells, and in particular cytotoxic CD8+ T cells, on the process of exercise-induced reduction of tumor growth. As the progression of the transgenic MMTV-PyMT model is predominantly regulated by myeloid cells (*Lin et al., 2003*), we carried out a further set of experiments, wherein immunocompetent FVB mice were given access to an exercise wheel 14 days prior to, and then following, subcutaneous injection with a murine mammary cancer line derived from the MMTV-PyMT model, the I3TC cell line (*Weiland et al., 2012*; *Figure 1A*). In this cell-line-driven model, allowing animals to exercise significantly reduced tumor growth and increased survival times (*Figure 1B*) relative to animals not given access to an exercise wheel.

To determine whether immune cell populations were also affected by exercise in the subcutaneous I3TC tumor model, we carried out flow cytometric analyses of single-cell suspensions of inoculated tumors, spleens, and tumor-draining lymph nodes of running versus non-running animals. Here, only CD8+ T cells showed a significant increase in frequency across these three tissues in the running animals (*Figure 1E*); no significant changes were seen in the frequency of CD4+ T cells, NK cells (*Figure 1C and D*), or intra-tumoral macrophages or neutrophils (*Figure 1—figure supplement 1I and J*).

To determine the role of increased cytotoxic T-cell populations in the reduction of tumor growth caused by exercise, we depleted CD8+ T-cells via weekly injections of anti-CD8 antibodies in both exercising and non-exercising mice during subcutaneous I3TC tumor growth. The CD8+ T-cell depletion reduced the CD8+ population of both the spleen (*Figure 1F*) and the tumor (*Figure 1G*), and significantly reduced the beneficial effects of exercise on both tumor growth (*Figure 1H*) and long-term survival (*Figure 1I*). This depletion study demonstrates a clear role for cytotoxic T-cells in the suppression of tumor growth by exercise in this model.

## Exercise alters the metabolic profile of plasma

Acute exercise induces changes in several known metabolic mediators of immune responses (*Henderson et al., 2004*). To investigate how the systemic availability of metabolites can change in response to exercise, we undertook a metabolomic investigation of muscle and plasma in response to exercise.

After habituation, an exhaustive endurance test was performed on wild-type mice on the FVB background, and following exercise, skeletal muscle and plasma were harvested immediately, snap frozen and analyzed by mass spectrometry (GC-MS) (*Figure 2—figure supplement 1A-C*).

Exercise affects a wide range of metabolic pathways; in particular, exercise reduced the frequency of glycolytic metabolites in skeletal muscle, for example, fructose-6-phosphate, glucose-6-phosphate and 3-phosphoglyceric acid (3 PG), while the frequency of several of the TCA metabolites, for example, citric acid, fumaric acid and malic acid, were significantly higher in muscle post-exercise (*Figure 2—figure supplement 1B*). This is in line with mitochondrial metabolism being rate-limiting in high-intensity exertion (*Brooks, 1998*). Interestingly, the TCA metabolites citric acid, malic acid and alpha ketoglutaric acid (aKG) were all higher in plasma after exertion (*Figure 2—figure supplement 1C*), suggesting release of these metabolites from muscle into plasma (*Figure 2—figure supplement 1D*). Contribution of muscle citrate to plasma has previously been shown; for example, by femoral arterio-venous sampling during exercise in human subjects (*Nielsen and Thomsen, 1979*; *Schranner et al., 2020*).

To extend the information to include the lymphoid organs, an additional set of exhaustive endurance tests was performed on mice followed by immediate harvest of skeletal muscle, plasma, spleen, and muscle draining (axillary) and non-draining (inguinal) lymph nodes. In addition, human plasma was harvested before (pre-exercise), directly after (post-exercise) and 1 hr (1 hr post) after an intense endurance exercise session in sedentary individuals (*Figure 2—figure supplement 1E*). These tissues were then analyzed by ultrahigh performance liquid chromatography/mass spectrometry (*Figure 2A*). As well as altering the intramuscular (*Figure 2B*) and circulating levels of the TCA metabolites (*Figure 2C and D*), acute exercise in these experiments also introduced shifts in the metabolic profiles of lymphoid organs, supporting the notion that acute exercise alters the metabolic environment of lymphoid organs (*Figure 2E–G*). Notably, the number of significantly increased metabolites was markedly higher in the muscle draining lymph nodes when compared to non-draining lymph nodes (*Figure 2F and G*, respectively). This indicates that changes seen in draining lymph

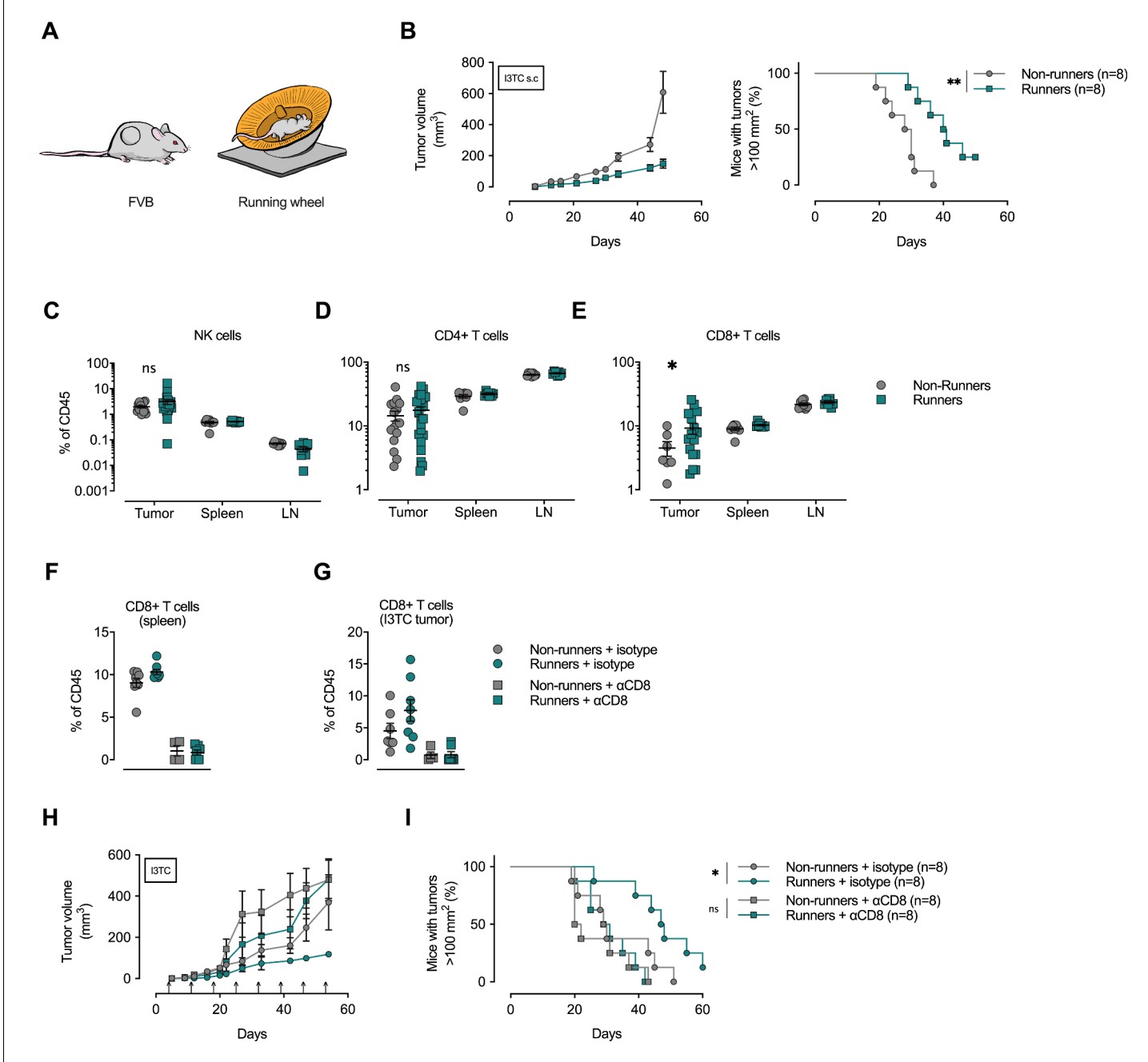

**Figure 1.** Depletion of CD8+ T-cells abolishes the anti-cancer effects of exercise training. (A) FVB mice were allowed to exercise voluntarily (in running wheels, runners) or left non-exercised (locked running wheels, non-runners) before and after being inoculated subcutaneously with $5 \times 10^5$ tumor cells of the breast cancer cell line I3TC. (B) Mean tumor volume and SEM over time (left) and survival (right). **p<0.01, *p<0.05, Log-rank (Mantel-Cox) survival test. (C–E) Flow cytometry determined frequency of lymphocytic populations within I3TC tumor, spleen and lymph nodes (LN) at day 55 after inoculation. *p<0.05 column factor combining all organs in a two-way ANOVA. ns = not significant. (F–G) Flow cytometry determined frequency of lymphocytic populations within spleen and I3TC tumors after CD8+ depletion and isotype control. (H–I) Same experimental setting as in (A) with (αCD8) or without (isotype ctrl) weekly antibody-mediated depletion of CD8+ T cells (arrows). Graphs show mean tumor volume and SEM over time (F) and survival (G). *p<0.05, Log-rank (Mantel-Cox) survival test.

The online version of this article includes the following figure supplement(s) for figure 1:

**Figure supplement 1.** Exercise trained MMTV-PyMT breast cancer mice show enhanced infiltration of GranzymeB+ cells.

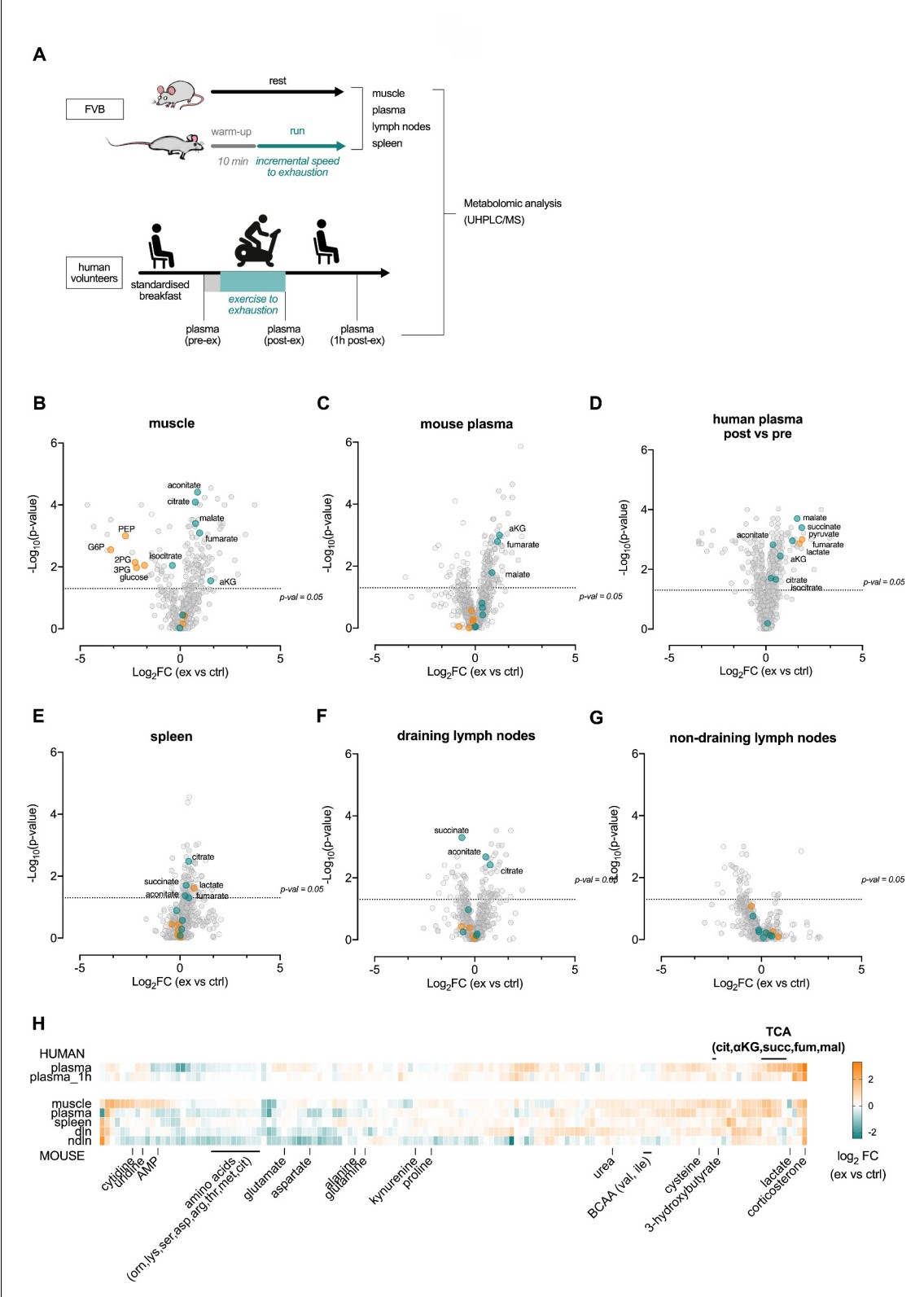

**Figure 2.** Acute exertion alters the metabolic profile of plasma and lymphoid organs. (**A**) FVB mice performed a single exhaustive treadmill exercise. Muscle, plasma, spleen, muscle draining, and non-draining lymph nodes were collected from exercising and control mice immediately after the exercise. Human subjects performed 30 min of endurance exercise at 75% of $W_{peak}$ and plasma samples were collected before (pre) and after (post) exercise. Samples were analyzed on the Precision Metabolomics mass spectrometry platform. Data is provided as *Figure 2—source data 1*. (**B–G**)

*Figure 2 continued on next page*

*Figure 2 continued*

Volcano plots of differentially induced metabolites per mouse tissue and human plasma. p-Values from the Student's T-test and the log$_2$FC (n = 6 and n = 8 for mouse and human samples respectively). TCA metabolites are colored blue. Glycolytic metabolites are colored orange. The dashed horizontal line is drawn at Y = -log$_{10}$(0.05). (**H**) Heatmap of exercise-induced changes in metabolite concentrations in human and mouse plasma and tissues. Metabolites with a significant change (adjusted p<0.05) in at least one tissue following exercise were included and clustered using hierarchical clustering.

The online version of this article includes the following source data and figure supplement(s) for figure 2:

**Source data 1.** Multiple organ metabolomics source data file.
**Figure supplement 1.** Acute exertion Increases the levels of TCA metabolites in muscle and plasma.
**Figure supplement 2.** Pathway analysis of post-exercise metabolic profiling.

nodes are likely attributable to muscle metabolite production. Interestingly, plasma, draining lymph nodes, and non-draining lymph nodes all showed higher levels of corticosterone after exercise (*Figure 2H*). Changes in amino acid and fatty acid levels were also identified in multiple organs (*Figure 2—figure supplement 1H*).

However, the single most profound metabolic change induced by exertion is the transient increase in circulating lactate. Circulating lactate levels rise very rapidly during exercise, and can (in response to high intensity exertion) increase up to 100-fold in skeletal muscle (*Bonen et al., 1998*; *Spriet et al., 1987*), with a concomitant increase of more than 10-fold in plasma (*Goodwin et al., 2007*). The rapid postmortem accumulation of systemic lactate in response to global oxygen deprivation made us unable to differentiate the lactate levels in murine organ samples (*Donaldson and Lamont, 2013*; *Keltanen et al., 2015*). An increase in lactate as well as TCA cycle metabolites was however seen in human plasma post-exercise (*Figure 2D*) and when measuring lactate in blood from the tail vein in live animals directly after an acute treadmill exercise (*Figure 2—figure supplement 1G*). In the human samples, these returned to close to resting levels 1 hr after exercise (*Figure 2—figure supplement 1F*), indicating that the changes in central carbon availability are likely conserved between mice and humans.

## Exercise-induced metabolites can modify activation of CD8+ T cells

As shown above, exercise can reduce tumor growth in a CD8+ T-cell-dependent manner. Given that metabolism and T-cell differentiation are tightly linked (*O'Neill et al., 2016*; *Pearce, 2010*), we next sought to determine whether any of the central carbon metabolites generated during exercise could be a determining factor in the action of CD8+ T cells in this process.

To test this, we activated CD8+ T cells ex vivo for 3 days in the presence of increasing doses (25 µM-50mM) of pH-neutralized lactic acid, pyruvic acid, citric acid, malic acid, succinic acid, fumaric acid, a-ketoglutaric acid, or sodium L-lactate. Reference human resting serum levels are provided in *Figure 3—figure supplement 1A*. Although the production of lactic acid is the immediate consequence of increased glycolytic flux during exercise, in healthy tissues, the increase in H+ is rapidly buffered in surrounding tissue and in plasma (*Péronnet and Aguilaniu, 2006*). For this reason, lactate produced during exercise does not significantly alter circulating plasma pH levels (unlike the acidification seen in, for example, solid tumors) (*Brand et al., 2016*). We therefore, adjusted pH to 7.35 for all metabolites prior to adding them to the cell culture media.

TCA metabolites can be transported across the plasma membrane with sodium carboxylate cotransporters (*Markovich, 2012*). Although none of the metabolites provided proliferative advantages after 3 days of activation, malate, succinate, and aKG were notably well tolerated by the T-cells at high doses (*Figure 3A*, *Figure 3—figure supplement 1B-C*). Activation in the presence of increasing concentrations of pyruvate and citrate reduced CD8+ T cell expansion (*Figure 3A Figure 3—figure supplement 1B-C*).

Malate, succinate, fumarate, aKG, and lactate all induced a loss of CD62L at around 1 mM concentrations, (*Figure 3B*, *Figure 3—figure supplement 1D*). Loss of CD62L surface expression is induced in response to TCR activation and instrumental for the migratory capacity of cells localized in secondary lymphoid organs.Based on the data from the metabolite screen, TCA metabolites appear to enhance the loss of CD62L in response to activation.

Furthermore, sodium L-lactate induced a dose-dependent increase in inducible T-cell costimulator (iCOS) and Granzyme B (GzmB) expression at day 3 of culture (*Figure 3C–D*, *Figure 3—figure*

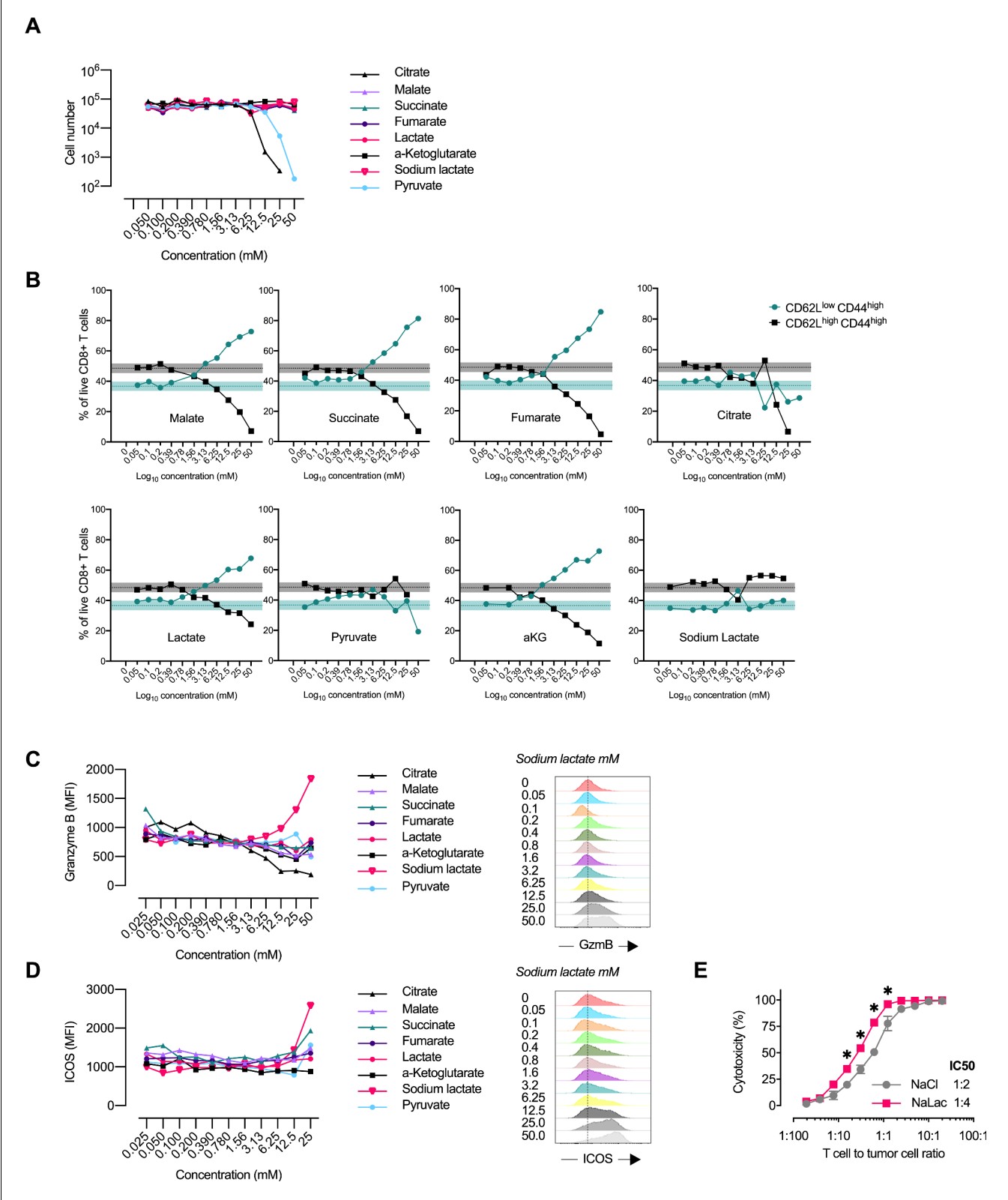

**Figure 3.** Central carbon metabolites alter the CD8+ T cell effector profile. (**A**) Proliferation of activated murine CD8[+] T cells in response to increasing concentrations of central carbon metabolites. The proliferation of activated murine CD8[+] T cells was assessed by using CountBright counting beads on flow cytometry at day 3 of culture. (**B**) Flow-cytometry-based CD62L and CD44 expression of live CD8[+] T cells expressed as frequency of cells at day 3 of culture with increasing concentrations of central carbon metabolites. Shaded areas represent mean and 95% confidence intervals at 0 mM. (**C**)

*Figure 3 continued on next page*

Figure 3 continued

Granzyme B median fluorescence intensity (MFI) at day 3 of culture with increasing concentrations of central carbon metabolites. Data from one representative experiment on CD8[+] T cells purified and activated from pooled spleens of multiple mice. (D) ICOS median fluorescence intensity (MFI) at day 3 of culture with increasing concentrations of central carbon metabolites. Data from one representative experiment on CD8[+] T cells purified and activated from pooled spleens of multiple mice. (E) Cytotoxicity against EL4-OVA tumor cells by OVA-specific OT-I CD8+ T cells activated for 3 days in the presence of 40 mM NaCl or NaLac. Graph represents specific cytotoxicity of n = 3 independent mouse donors at varying effector-to-target ratios. *p<0.01 repeated-measures two-way ANOVA with Sidak's multiple comparison test.

The online version of this article includes the following figure supplement(s) for figure 3:

**Figure supplement 1.** Profiling of CD8+ T cell responses to central carbon metabolites.

---

supplement 1E-F). We found no significant effect of metabolite exposure on T-cell CTLA-4 expression (*Figure 3—figure supplement 1G*). The effect of sodium L-lactate could be detected starting at concentrations of approximately 6 mM. ICOS is expressed on activated T-cells and GzmB is the most highly expressed effector protein in activated mouse CD8+ T cells (*Hukelmann et al., 2016*) and is essential to granule-mediated apoptosis of target cells (*Heusel et al., 1994*). In keeping with this, in co-culture with ovalbumin (OVA) expressing EL4 tumor cells, OVA-specific OT-I CD8+ T-cells activated in the presence of sodium L-lactate for 72 hr showed increased cytotoxicity against the tumor target cells (*Figure 3E*).

## Acute exercise alters CD8+ T cell metabolism in vivo

Because of our findings of altered T-cell function in response to exercise-induced metabolites, we next sought to investigate if there is evidence of an alteration in the CD8+ T-cell metabolome in vivo after intense exertion. To address if exercise can alter the central carbon metabolism of activated CD8+ T cells in vivo, we employed an OVA vaccination model and adoptive transfer of transgenic, OVA-specific OT-1 CD8+ T-cells where the timing of the T-cell activation can be controlled. In short, transgenic OT-I CD8+ CD45.1+ T-cells were administered to recipient animals, followed by vaccination with bone marrow derived macrophages (BMDM) presenting ovalbumin to activate the OT-1 T-cells. Two or 3 days after the vaccination, a bolus of [U-$^{13}$C$_6$]glucose was introduced to resting and exercising mice (*Figure 4A*), and the central carbon metabolism of the OT-1 CD8+ T-cells isolated from the spleen was assayed. The [U-$^{13}$C$_6$]glucose was introduced after warm up on a treadmill for 10 min at low speed, so as to ensure maximal glucose uptake by the skeletal muscle at the time of injection. The spleens were harvested at 20 min post-exercise and at the equivalent time point in the resting animals. The data shows a diverging carbon metabolism of the exercised CD8+ T-cells at the level of m+three labeled pyruvate and m+two labeled aKG, indicating an altered enzymatic activity or contribution of extracellular labeled molecules (*Figure 4B and C*), providing evidence that exercise can alter the metabolism of CD8+ T-cells in vivo.

## Target-specific CD8+ T cells transferred from trained mice show enhanced antitumoral capacity

In order to determine the functional effects of exercise on the CD8+ T-cell population, adoptive transfer of naïve CD8+ T-cells from exercising OT-1 animals was carried out. These T-cells were transferred to C57Bl6 inbred strain recipient mice that had been inoculated with an OVA-expressing melanoma (B16-F10-OVA) (*Figure 5A and B*). Following the transfer of T-cells to non-exercising mice, tumor growth was monitored for 40 days. Blood profiles on day 10 following transfer confirmed the expansion of the OT-1 population in the recipient mice, and also showed a significant increase in expression of iCOS in the cells transferred from exercising donors (*Figure 5C and D*). The sedentary recipient animals that received T-cells from exercising donors showed an enhanced survival and reduced rate of tumor growth, when compared to sedentary animals receiving T-cells from sedentary donors (*Figure 5E–G*). This indicates that there is a persistent and positive effect on the efficacy of anti-tumor CD8+ T-cells when the T-cells are derived from exercising animals.

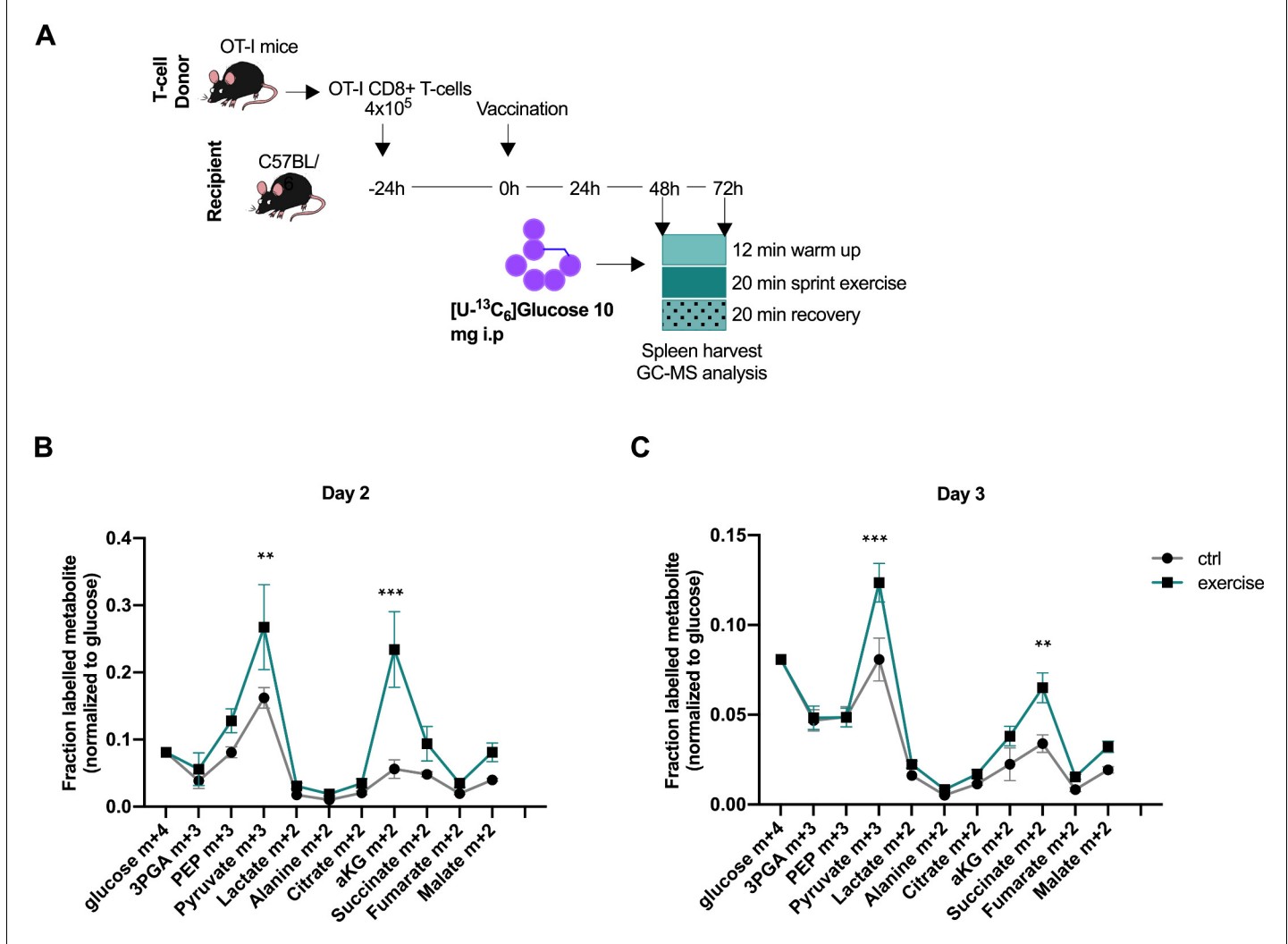

**Figure 4.** Acute exercise alters CD8+ T cell metabolism in vivo. (**A**) Recipient mice received OT-I CD8+ T-cells followed by vaccination. On day 2 and 3 after vaccination, 10 mg of [U-$^{13}C_6$]glucose was introduced to the mice prior to a treadmill exercise. CD8+ T-cells were harvested from the spleen and incorporation of labeled carbons in cellular metabolites was assessed by GC-MS. Data is provided as *Figure 4—source data 1*. (**B**) Fraction of labeled metabolites in CD8+ T-cells 48 hr after vaccination. (**C**) Fraction of labeled metabolites in CD8+ T-cells 72 hr after vaccination. **$p<0.01$ and ***$p<0.001$ using a two-way ANOVA with Sidak's multiple comparison test.

The online version of this article includes the following source data and figure supplement(s) for figure 4:

**Source data 1.** Glucose derived carbon distribution in CD8+ T cells source data file.

**Figure supplement 1.** Glucose levels in CD8+ T cells of exercising mice.

**Figure supplement 2.** Exercise effects on glucose derived carbon distribution in CD8+ T cells.

## Daily administration of sodium L-lactate delays tumor growth in a CD8+ T-cell-dependent manner in vivo

Given the above results that exercise-induced metabolites can alter the properties of immune cells, we wanted to investigate if the lactate-induced increases in CD8+ T cell differentiation markers and

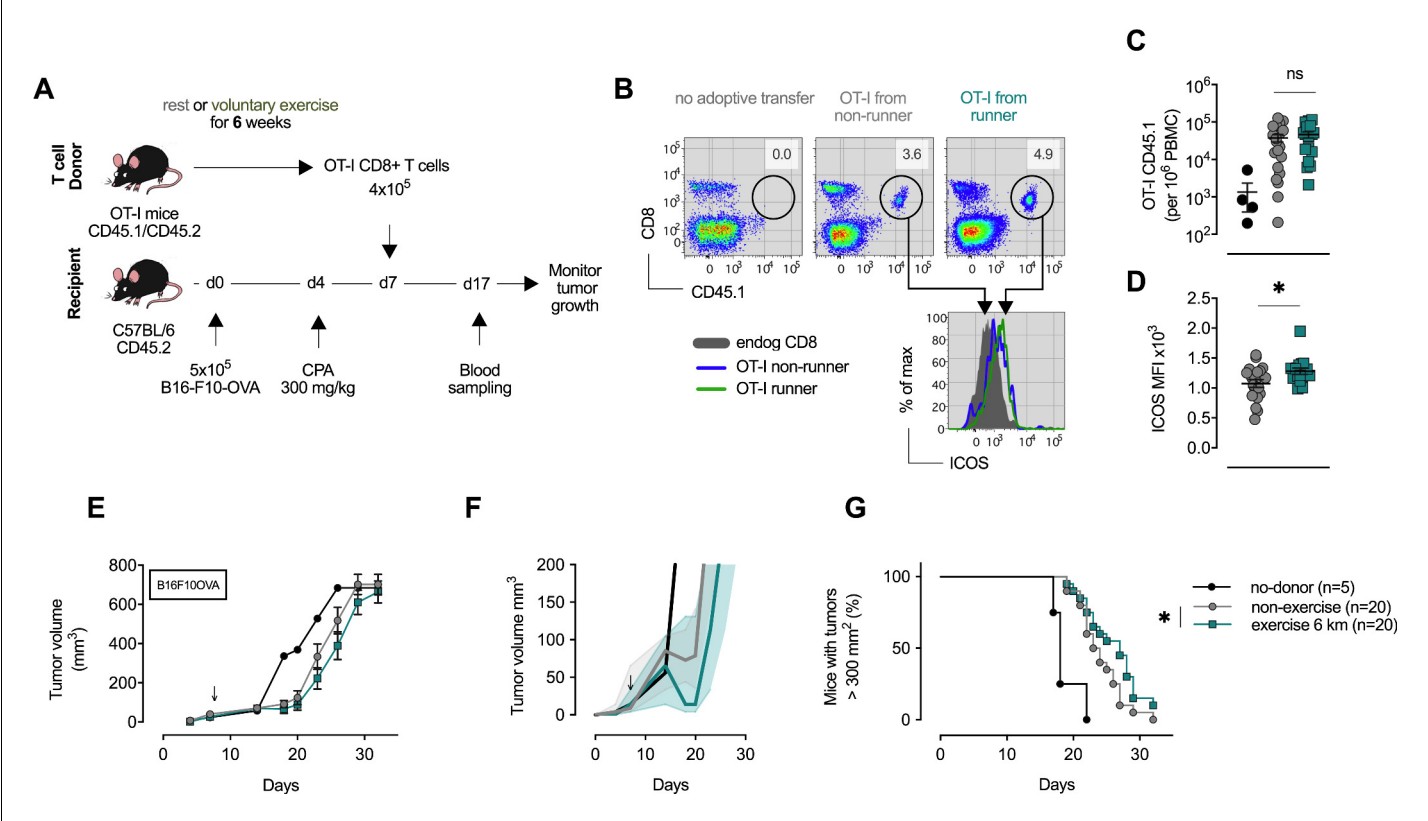

**Figure 5.** CD8+ T cells transferred from trained mice show enhanced anti-tumoral capacity. (**A**) OT-I mice carrying the congenic marker CD45.1 were given access to a locked or moving running wheel for 6 weeks. In parallel, C57Bl/6 (CD45.2) animals were inoculated with $5 \times 10^5$ ovalbumin (OVA)-expressing B16-F10 (B16-F10-OVA) melanoma and conditioned with 300 mg/kg cyclophosphamide (CPA). $4 \times 10^5$ OVA-reactive OT-I CD8+ T cells were isolated from running or non-running mice and adoptively transferred into tumor-bearing animals. Peripheral blood was sampled 10 days after adoptive transfer and tumor volume monitored. (**B**) Flow cytometry analysis of OT-I T cell expansion in peripheral blood 10 days after adoptive transfer. Adoptive cells were distinguished from endogenous immune cells by expression of the CD45.1 congenic marker. Histogram shows ICOS surface expression on adoptive and endogenous CD8+ T cells. (**C**) Frequency of adoptive OT-I T cells in peripheral blood. *p<0.05, two-tailed t-test. ns = not significant. (**D**) Median fluorescence intensity (MFI) of ICOS (bottom) of adoptive OT-I T cells in peripheral blood. *p<0.05, two-tailed t-test. ns = not significant. (**E–G**) Mean tumor volume and SEM (**E**), median IQR (**F**) and survival (**G**) over time. ↓ depicts time of OT-I T cell injection *p<0.05, Log-rank (Mantel-Cox) survival test.

The online version of this article includes the following figure supplement(s) for figure 5:

**Figure supplement 1.** OT-I mice carrying the congenic marker CD45.1 were given access to a locked or moving running wheel for 6 weeks.

cytotoxic efficacy might extrapolate to affect tumor growth in vivo. Therefore, we performed daily infusions of sodium L-lactate into tumor-bearing animals at doses that result in plasma lactate levels similar to those seen during intensive exercise (approximately 10–20 mM). As can be seen in *Figure 6—figure supplement 1A*, an intraperitoneal injection of a 2 g/kg dose of sodium L-lactate results in a 18 mM spike in serum lactate concentrations at 20 min post-injection. Following this dose, levels subside to 4 mM within 60 min; the expected time to reach baseline values from this magnitude of spike is approximately 180 min post-injection (*Lezi et al., 2013*). This dose was chosen as an approximation of the levels and persistence of rises in plasma lactate that occur following intense short-term periods of exercise.

Female FVB mice, given daily doses of 2 g/kg sodium L-lactate, showed a decrease in overall tumor growth after inoculations with the I3TC tumor cell line (*Figure 6A*). Similar results were obtained with the colon adenocarcinoma MC38 cell line on C57BL/6 animals (*Figure 6C*), with accompanying increases in tumor-bearing animal survival. Lower doses of lactate (0.5–1 g/kg) did not significantly alter tumor growth (*Figure 6A*), while a higher daily dose of Sodium L-lactate, 3 g/ kg, caused a significant reduction in tumor growth, with approximately the same efficacy as the 2 g/

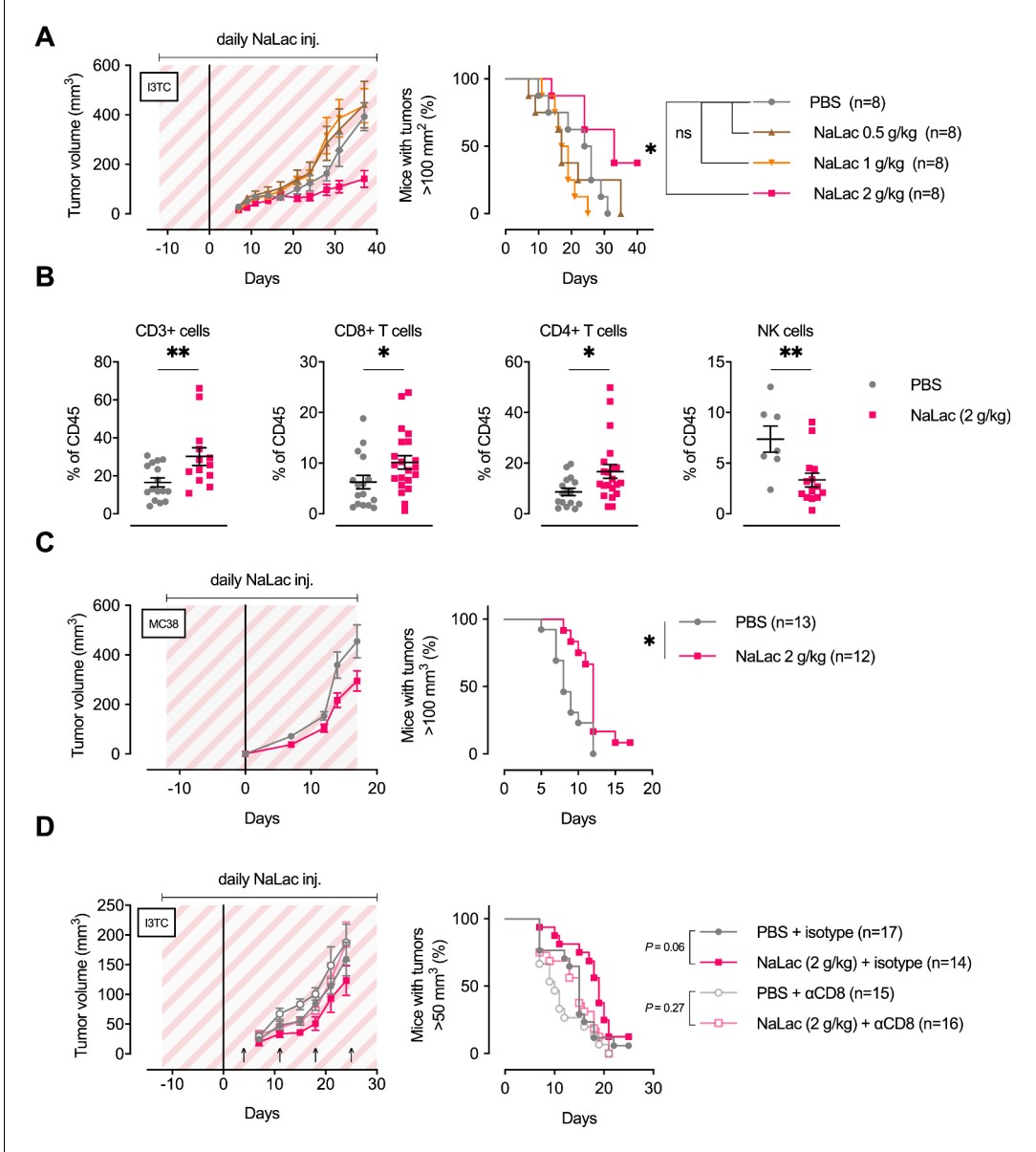

**Figure 6.** Daily administration of sodium L-lactate delays tumor growth in vivo. (**A**) Daily doses of PBS or 0.5, 1, or 2 g/kg Sodium L-lactate (NaLac) were administered i.p to FVB mice for 12 days before subcutaneous inoculation with $5 \times 10^5$ cells of the MMTV-PyMT-derived breast cancer cell line I3TC. Daily sodium L-lactate injections were continued throughout the experiment. Graphs show tumor volume (mean and SEM) over time and survival. *p<0.05, Log-rank (Mantel-Cox) survival test. (**B**) Flow cytometric characterization of I3TC tumor infiltrating immune cell populations. **p<0.01, *p<0.05, two-tailed t test. (**C**) Daily doses of PBS or 2 g/kg NaLac was administered i.p to C57BL/6J mice for 12 days before subcutaneous inoculation with $5 \times 10^5$ cells of the colon cancer cell line MC38. Injections were continued throughout the experiment. Graphs show tumor volume (mean and SEM) over time and survival. **p<0.01, Log-rank (Mantel-Cox) survival test. (**D**) Same experimental setting as in (**A**) with or without weekly antibody-mediated depletion of CD8+ T cells (arrows). Graphs show tumor volume (mean and SEM) over time and survival. ns = not significant, Log-rank (Mantel-Cox) survival test.

The online version of this article includes the following figure supplement(s) for figure 6:

**Figure supplement 1.** Characterization of tumor infiltrating CD8+ T cells after daily administration of sodium L-lactate.

kg dose (*Figure 6—figure supplement 1B*). As shown in *Figure 6B*, there were significant changes in tumor infiltrating immune populations in animals treated with 2 g/kg of lactate, namely, increases in the frequency of total (CD3+) T cells, including both CD4+ and CD8+ T cells. The frequency of tumor-infiltrating NK cells was reduced. Despite finding increased numbers of tumor infiltrating CD8

+ T cells (*Figure 6B* and *Figure 6—figure supplement 1C*), daily lactate administration did not change the percentage of GzmB+ cells or the GzmB, PD1 or CTLA intensity on CD8+ infiltrating cells (*Figure 6—figure supplement 1D-E*).

To determine whether the effect of lactate infusion is dependent on CD8+ T-cell-mediated effects on tumor growth, we depleted the cytotoxic T cell population via injection of an anti-CD8 antibody (*Figure 6D*) and observed that the net effect of lactate injection was reduced to statistical insignificance by the depletion of the CD8+ T cell population. This indicates that the effect of daily lactate injections on moderating tumor growth is mediated by CD8+ T cells.

## Discussion

Here, we show that exercise requires cytotoxic T cells to affect tumor growth. Recent studies suggest that exercise reduces cancer recurrence and mortality, and the effect of exercise on tumorigenic progression has now been documented in a range of animal models (*Ruiz-Casado et al., 2017*). Previously proposed underlying mechanisms for the anti-neoplastic effects of exercise include effects on weight control and endocrine hormone levels, as well as altered tumor vascularization (*Betof et al., 2015*) and immune function (*Gleeson et al., 2011*). Exercise has been shown to reduce systemic inflammation (*Gleeson et al., 2011*), cause increases in circulating numbers of immune cells (*Gustafson et al., 2017*) and enhance vaccination response (*Ledo et al., 2020*). Systemic levels of both CD8+ T cell and NK cell levels increase transiently in response to acute exercise. There is evidence that these immune cells populations exhibit an effector phenotype (*Campbell et al., 2009*) and an increased peripheral tissue homing capacity (*Krüger and Mooren, 2007*), both important for anti-tumor activity.

Exercise is a multimodal stimulus. Systemic signaling from activated skeletal muscle to immune cell function has been attributed previously to for example skeletal muscle-derived myokines such as IL-6 and Oncostatin-M (OSM), and to catecholamine release (*Gleeson et al., 2011*). In a recent study, Hojman et al., showed that an increase in systemic levels of epinephrine during exercise together with skeletal muscle IL-6 could increase NK cell recruitment during tumorigenesis (*Pedersen et al., 2016*). However, in the mouse models employed here we found limited infiltration and no effect of exercise on the CD3- NK1.1+ population, relative to our observation of an increased frequency of CD3+/CD8+ T cells. The results from depletion and adoptive transfer of the CD8+ population provide further support for the importance of cytotoxic T-cells in mediating the anti-neoplastic effects of exercise.

Immune cells patrol all corners of the body and can reside in niches with highly differential metabolic profiles. Recent findings have provided evidence that these different environments can instruct effector functions of immune cells (*Buck et al., 2017*). In the current study, we show how exercise modulates metabolic parameters significantly beyond local skeletal muscle. We found that lactate and TCA metabolites accumulated in both the active skeletal muscle and in plasma and secondary lymphoid organs after acute exertion.

As trafficking immune cells circulate, they will often be exposed to high levels of accumulating metabolites, in both loci of disease and in metabolically active tissues. However, the principal shaping of immune function happens at the point of activation, primarily occurring in the lymphoid organs. In evaluating the role an altered metabolic environment plays in the process of modifying CD8+ T-cell function, we found that key metabolites, including lactate, had unexpected functions in the activation of CD8+ T cells, with multiple TCA metabolites stimulating the loss of CD62L. CD62L shedding from TCR activated CD8+ T-cells has been suggested to be mTOR and PI3K dependent (*Sinclair et al., 2008*) and aKG-mediated activation of mTORC1 has been reported in multiple cell lines (*DeBerardinis et al., 2007*; *Durán et al., 2013*; *Villar et al., 2015*), as well as in activated CD8 + T-cells (*Suzuki et al., 2018*), suggesting that the metabolite-induced loss of CD62L may be mediated through mTOR activation. In addition, we found that lactate induces an effector profile and an overall increase in cytotoxic activity.

In contrast to our findings, within the local micro-environment, tumor-derived lactate has been shown to inhibit the antitumor functions of both T and NK cells due to intracellular acidification (*Brand et al., 2016*). Likewise, lactate-associated acidity has been shown to impair CD8+ T cell motility (*Haas et al., 2015*). Importantly, within poorly vascularized and fast-growing tumor masses, or in laboratory ex vivo cultures, lactic acid export from cells results in a significant acidosis.

However, healthy tissue microenvironments are very efficiently buffered, and capable of maintaining a physiological pH range even when large amounts of lactate are produced.

Our findings that circulating levels of TCA metabolites increase by 2–8 fold after exercise suggests that at peripheral sites, substantial amounts of these metabolites have been released into the lymphatic fluid draining the muscle tissue. Additionally, a recent paper using lymphatic cannulation showed that TCA metabolites are readily available in, for example, mesenteric lymph (*Zawieja et al., 2019*). Thus, to mimic the activation of T cells in a lymphoid organ exposed to the physiologically buffered metabolites generated by exercising muscle, we activated CD8+ T cells in the presence of TCA metabolites and lactate in media adjusted to a physiological pH.

The use of distally produced lactate as a fuel for oxidative respiration was described for inactive muscle already in 1970 (*Jorfeldt, 1970*) and has recently been reported in both malignant and non-malignant cell types (*Faubert et al., 2017*; *Hui et al., 2017*), suggesting that lactate can be used as a primary fuel by CD8+ T cells, at least during early activation. In vivo administration of labeled glucose two and three days after vaccination, led to differential levels of labeled pyruvate and alpha-ketoglutarate in CD8+ T-cells of the spleen post-exercise. Dilution of 13C labeling between glycolysis and TCA cycle intermediates can imply oxidation of alternative fuels (*Hensley et al., 2016*). The administered 13C-glucose was, as expected, used as a carbon source by CD8+ T cells. Although total glucose levels were similar between T-cells from exercising and non-exercising animals (*Figure 4—figure supplement 1A*), the availability of labeled glucose appeared lower in exercised animals, reflected by lower total 13C-glucose counts (*Figure 4—figure supplement 1B*). The reduced availability is most likely due to dilution of the 13C-glucose in the circulating blood glucose pool due to a contribution from hepatic glycogenolysis, stimulated by the increased metabolic demand during exercise. When normalized to the labeled glucose fraction, exercise altered the central carbon metabolism of activated CD8+T-cells in vivo at the level of m+3 labeled pyruvate and m+2 labeled aKG. This finding suggests an increased uptake of systemic metabolites by activated T-cells during exercise.

When isolating the exercise effect on the CD8+ T-cell population by transferring cells from exercised animals to tumor bearing, non-exercising recipients, we show an enhanced efficiency in reducing tumor growth by the T-cells obtained from exercising individuals. These results demonstrate that in our models, the exercise effect is exerted by and inherent to the CD8+ T cell population.

In the experimental setting where animals were given daily doses of high levels of sodium L-lactate, we found an increase in intratumoral CD8+ T-cells and a decrease in overall tumor growth. In addition, we found high levels of CD4+ cells and reduced NK-cell infiltration, supporting previous evidence that lactate can affect multiple cells of the immune system. These findings indicate that lactate infusion mimics some of the effects of exercise, but that exercise has additional, integrative, components beyond merely increased levels of lactate.

In conclusion, we have shown that the antitumoral effects of exercise depend on CD8+ T-cells, and that intense exertion can alter the intrinsic metabolism and antitumoral effector function of cytotoxic T-cells. This indicates that, in the physiological setting, exercise-derived metabolites, whether systemically delivered or draining into an adjacent lymph node, could act to boost a nascent T cell response. This indicates that the adaptive immune system is a key component of exercise-induced suppression of tumorigenesis.

## Materials and methods

**Key resources table**

| Reagent type (species) or resource | Designation | Source or reference | Identifiers | Additional information |
|---|---|---|---|---|
| Strain, strain background (*M. musculus*) | FVB/N | Janvier Labs | FVB/N | FVB 'WT' mice |
| Strain, strain background (*M. musculus*) | C57Bl/6J | Charles River | C57Bl/6J (JAX) | C57Bl/6J 'WT' mice |

*Continued on next page*

*Continued*

| Reagent type (species) or resource | Designation | Source or reference | Identifiers | Additional information |
|---|---|---|---|---|
| Strain, strain background (*M. musculus*) | OT-I | The Jackson Laboratory (JAX) | Cat 003831 | TCR transgenic mice |
| Strain, strain background (*M. musculus*) | CD45.1 | The Jackson Laboratory (JAX) | Cat 002014 | CD45.1 congenic marker |
| Cell line (*M. musculus*) | I3TC | *Weiland et al., 2012* | | Murine breast cancer cell line derived from MMTV-PyMT FVB mice Mycoplasma test data RATIOS under 1 = clear.I3TC 0.3381995134 |
| Cell line (*M. musculus*) | B16F10 | ATCC | CRL-6475 | Mouse melanoma cell line Mycoplasma test data RATIOS under 1 = clear. B16 OVA 447 163 0.36465324 |
| Cell line (*M. musculus*) | LLC | ATCC | CRL-1642 | Mouse lung carcinoma cell line Mycoplasma test data RATIOS under 1 = clear. LLC 438 151 0.34474885 |
| Antibody | aSINFENKL | BioLegend | Clone 25-D1.16 | |
| Chemical compound, drug | Sodium L-lactate | Sigma | L7022 | Metabolite |
| Chemical compound, drug | Cyclophosphamide | Sigma | C0768 | Chemotherapy agent |
| Commercial assay or kit | CD8a microbeads | Miltenyi Biotec | 130-117-044 | Beads for magnetic sorting |
| Chemical compound, drug | Collagenase A | Fisher Scientific | 50-100-3278 | For single cell dissociation of tumors |
| Chemical compound, drug | DNase | Sigma | D5025 | For single cell dissociation of tumors |
| Commercial assay or kit | Dynabeads Mouse T-activator | Thermo Fisher | 11456D | Dynabeads Mouse T-activator CD3/CD28 |
| Chemical compound, drug | IL-2 | Sigma | 11011456001 | IL-2 |
| Chemical compound, drug | Citric acid | Sigma | 251275 | |
| Chemical compound, drug | Malic acid | Sigma | M1000 | |
| Chemical compound, drug | Succinic acid | Sigma | S3674 | |
| Chemical compound, drug | Fumaric acid | Sigma | 47910 | |
| Chemical compound, drug | Lactic acid | Sigma | L1750 | |
| Chemical compound, drug | Pyruvic acid | Sigma | 107360 | |
| Chemical compound, drug | a-Ketoglutaric acid | Sigma | K1750 | |
| Chemical compound, drug | Oxaloacetic acid | Sigma | O4126 | |
| Commercial assay or kit | Resazurin reduction assay | Sigma | R7017 | T-cell proliferation assessment |
| Commercial assay or kit | Dead cell stain kit | Thermo Fisher | L10119 | Fixable Near-IR Dead cell Stain Kit |

*Continued on next page*

Continued

| Reagent type (species) or resource | Designation | Source or reference | Identifiers | Additional information |
|---|---|---|---|---|
| Antibody | aCD44 (rat monoclonal) | BD Biosciences | Clone IM7 | 1:2000 |
| Antibody | aCD45.1 (mouse monoclonal) | BD Biosciences | Clone A20 | 1:200 |
| Antibody | aCD45.2 (mouse monoclonal) | BD Biosciences | Clone 104 | 1:200 |
| Antibody | CD8 (rat monoclonal) | BD Biosciences | Clone 53–6.7 | 1:200 |
| Antibody | aCTLA-4 (Armenian hamster monoclonal) | BD Biosciences | Clone UC10-4F10-11 | 1:200 |
| Antibody | aCD62L (mouse monoclonal) | BioLegend | Clone MEL-14 | 1:200 |
| Antibody | aICOS (Armenian hamster monoclonal) | BioLegend | Clone C398.4A | 1:200 |
| Antibody | aGzmB (mouse monoclonal) | Thermo Fisher | Clone GB12 | 1:100 |
| Commercial assay or kit | Counting beads | Thermo Fisher | C36950 | CountBright Absolute Counting Beads |
| Software | FlowJo | Tree Star | Version 8.8.7 | |
| Software | Prism | GraphPad | Version 8 | |
| Software | R | R Core Team | Version 3.6.1 | |
| Antibody | CD3 (rabbit polyclonal) | Abcam | #ab5690 | IHC 1:100 |
| Antibody | F4/80 (rat monoclonal) | Abd Serotec | #MCA497 | IHC 1:100 |
| Antibody | Podocalyxin (goat polyclonal) | R and D Systems | #AF1556 | IHC 1:200 |
| Antibody | PCNA (mouse monoclonal) | Dako | #M0879 | IHC 1:500 |
| Antibody | GzmB (rabbit polyclonal) | Abcam | #ab4059 | IHC 1:100 |
| Commercial assay or kit | ABC kit | Vector | PK6100 | Avidin-Biotin peroxidase Complex |
| Commercial assay or kit | DAB | Vector | SK4100 | Diaminobenzidine staining |
| Software | Image J | NIH | | |

## Animals

All experiments and protocols were approved by the regional animal ethics committee of Northern Stockholm (dnr N78/15, N101/16). Female wild type (WT) FVB-mice were bred with transgenic male mice carrying the PyMT oncogene under the MMTV promotor (*Lin et al., 2003*) on the FVB inbred strain background. The female offspring, being either PyMT-positive (heterozygous) or WT, were from an age of 4 weeks introduced to voluntary running conditions. For sprint and inoculation experiments, female wild type (WT) FVB-mice, 6–7 weeks of age were purchased from Janvier Labs (France), allowed 7 days of acclimatization and housed two mice per cage in a temperature controlled room (20 ± 2°C) with dark-light cycles of 12 hr and access to food and water ad libitum. For T cell purification, and adoptive transfer, wild type donor and recipient C57BL/6J animals were purchased from Janvier Labs. TCR-transgenic OT-I mice (catalogue 003831, The Jackson Laboratory) were crossed with mice bearing the CD45.1 congenic marker (catalogue 002014, The Jackson Laboratory).

## Animal exhaustion exercise

The mice were habituated to the treadmill (Columbus instruments OHIO Exer 3/6) for 4 to 5 days prior to the test. Habituation was performed by first placing the mice on a stationary band, with a 10° uphill slope, for a few minutes. The speed was then gradually increased for about 1 min, to a maximum of 6 m/min, which was then kept for 5 min. On the following days, the same procedure was performed, and time and speed were gradually increased in order to ensure that the mice were fully familiarized with running on the treadmill. During the test, the mice ran at 10° uphill. An initial 10 min warm-up (10 m·min− 1) was followed by gradual increases of 2 m·min− 1 every 2 min. Exhaustion was determined as the time when the mouse had withstood three mild motivations without attempts to continue running. Control mice were kept in the cage during the time of the test.

## In vivo exercise and tumor growth

Four-week-old PyMT+ females of the MMTV-PyMT strain were housed individually in a temperature controlled room ($20 \pm 2°C$) with 12 hr dark-light cycles and food and water ad libitum. The mice were allowed free access to a running wheel (Med Associates Inc ENV-044, St Albans, USA). Running patterns and distance were monitored wirelessly using appropriate software (Med Associates Inc SOF-860 Wheel Manager and Med Associates Inc SOF-861 Wheel Analysis). A control group with identical but locked wheels was included in the study to control for environmental enrichment. Animal body weight was monitored weekly throughout the experiment and mammary gland tumors measured twice weekly using calipers. Tumor volumes (V) were approximated using the formula $V = (\pi/6) * l * w * h$. Experiments were terminated when the animals reached 12 weeks of age.

FVB WT animals were introduced to running or locked wheels and allowed to acclimate to running for 2 weeks prior to injections of $5 \times 10^5$ tumor cells of the I3TC cell line. The cells were trypsinized, washed once with PBS and suspended in 100 µL sterile PBS and injected subcutaneous (s.c) into running and non-running FVB WT mice. Animal body weight was monitored throughout the experiment. Tumors size was measured as previously described. Experiments were terminated 8 weeks after tumor cell injections. Upon termination of the experiments, tumor specimens were surgically removed and fixed in 4% paraformaldehyde (Sigma, F8775) or dissociated for single cells suspension.

For CD8 depletion, 200 µg of antibodies (anti-mCD8α clone 53–6.72, BioXCell or Rat IgG2a Isotype control, clone 2A3, BioXCell) was injected i.p at day 4 post-tumor cell injections and repeated weekly throughout the experiment.

For T-cell adoptive transfer, 8 to 12 weeks old female C57BL/6J were inoculated sub-cutaneously with $5 \times 10^5$ B16-F10-OVA and conditioned 4 days later with peritoneal injection of 300 mg/kg cyclophosphamide (Sigma, C0768) in PBS. At day 7, $4 \times 10^5$ transgenic OT-I from running/non-running mice were injected peritoneally to the tumor bearing recipient mice. Tumor volume was measured every 2–3 days with electronic calipers until day 60. At day 17 after OT-I cell transfer, peripheral blood was collected from tail vein, red blood cells lysed with water, and leukocyte cell suspension analyzed by flow cytometry.

## In vivo lactate and tumor growth

To evaluate the effect of lactate on tumor growth in vivo, FVB and C57/Bl6J WT animals were subject to daily intraperitoneal (i.p) Sodium L-Lactate (Sigma, L7022) injections (0.5, 1, 2, or 3 g/kg) for 12 days prior to injections of $5 \times 10^5$ tumor cells of the I3TC or MC38 cell line. Tumor growth was monitored as previously described and daily injections of Sodium L-Lactate was continued throughout the experiment. To ascertain transient levels of sodium L-lactate in plasma induced by lactate injection, 2 g/kg of Sodium L-Lactate was administered i.p to FVB and C57Bl6 animals (n = 1), systemic levels of lactate were monitored through tail vein bleedings pre-injection, and at 5,10, 20, 40, and 60 min post-injection (AccutrendPlus).

## Human acute exercise

Eight healthy, non-smoking sedentary men aged 34–51 were recruited after filling in a questionnaire to assess their health status and performing a peak oxygen uptake test ($\dot{V}O_2$ peak test) to assess their maximal performance level. All subjects completed a single bout of acute endurance exercise which consisted of 30 min of cycling at 75% of their $W_{peak}$. Food intake was controlled for by

providing a standardized breakfast. Blood was sampled from V. mediana cubiti at three timepoints: immediately pre and 1 and 3 hr following acute endurance exercise using Li-Heparin Plasma separation tubes (BD Vacutainer #367377, BD Biosciences). Plasma was separated by centrifugation at 3000 g for 10 min and immediately frozen at −80˚C.

## In vivo 13C Glucose test

For the in vivo 13C glucose test, 8 to 12 weeks old female C57BL/6J mice were habituated to the treadmill. On day −1, $2 \times 10^6$ transgenic OT-I-CD8+ T-cells were injected intra peritoneally. On day 0, the mice were vaccinated using OVA-antigen presenting BMDMs, and on days 2 and 3, the mice were split into running and resting mice (n = 4). The running mice were allowed to warm up on the treadmill, before 10 mg of [U-$^{13}$C$_6$] glucose was injected peritioneally and the running mice performed the 20 min incremental endurance test, as previously described. The mice were then allowed to recover for 20 min before cervical dislocation euthanasia and collection of organs. Spleens were harvested and placed in ice-cold PBS on ice and quadriceps muscle from one hind leg were dissected, frozen in liq $N_2$ and stored at −80 ˚C until further processing. $2 \times 10^6$ CD45.1+ CD8+ splenocytes were isolated and frozen on a dry ice and ethanol slurry and stored at −80 ˚C until further processing.

## Cell lines

EL4 was a gift from Prof. H. Stauss (UCL, London). B16-F10 and LLC were purchased from ATCC (CRL-6475 and CRL-1642, respectively). I3TC was originally derived from the FVB MMTV-PyMT breast cancer model (Weiland et al., 2012).

## Generation of ovalbumin-expressing cell lines

B16-F10, LLC and cells were co-transfected with the transposon vector pT2 encoding OVA, eGFP and neomycin phosphotransferase and the vector encoding transposase SB11. Three days later 400 mg/ml G418 (Gibco, 10131035) was added to culture media to select for transgene-expressing cells. Successful integration was confirmed by analyzing eGFP fluorescence by flow cytometry. Limiting dilution was used to derive monoclonal OVA-expressing lines for each cell line. OVA presentation was confirmed by flow cytometry using a PE-labeled antibody against surface SIINFEKL bound to H-2Kb (clone 25-D1.16, BioLegend).

## Generation of antigen-presenting dendritic cells

Mouse femur and tibia were isolated from sacrificed mice. After sterilizing in ethanol and transferring to sterile PBS, muscle tissue was removed and tibia separated from femur. To isolate the bone marrow, bones were trimmed at both sides and flushed with 10 mL of sterile PBS to retrieve bone-marrow-derived cells. These were pelleted by 5 min centrifugation at 200 rcf and resuspended in 1 mL ACK lysis buffer for 2 min to lyse red blood cells. The reaction was stopped using 40 mL PBS and cells pelleted as before, and resuspended in BMDM media (DMEM, 10% FBS, 1% PS, 10 ng GM-CSF, 10 ng M-CSF). After plating on 10 cm cell culture dishes, cells were cultured for 7 days; GM-CSF and M-CSF was replenished every 2 days. At day 7, BMDM media was removed and replaced with RPMI (with glutamine) + 100 ng/mL LPS (to activate BMDMs for antigen presentation by inducing MHC, CD80, and CD86 expression), followed by 24 hr incubation. Next, BMDMs were lifted using 4 mL of Corning cell stripper (Corning, Catalog #25–056 CI) along with a cell lifter. After stopping the reaction in 8 mL RPMI media, cells were spun down, the pellet resuspended in pure RPMI, and counted. To this mixture, SIINFEKL (an OVA fragment that can be presented by mouse MHC class I molecule H2k$^b$) was added to a final concentration of 100 ng/mL and cells incubated for 1 hr at 37 degrees Celsius, with shaking every 15 min to prevent attachment. two washes with PBS preceded resuspension in PBS at a concentration of $10*10^6$ cells/mL for injection.

## Generation of OVA specific OT-I transgenic CD8+ T- cells

Spleens and lymph nodes were obtained from OT-1 transgenic mice (Jackson laboratory #003831) and kept in PBS. Using a 40 µM strainer on a 50-mL tube, spleens and lymph nodes were filtered through the strainer using the flat end of a 1-mL syringe. This solution was spun at 200 rcf for 5 min to pellet cells. After discarding the supernatant, 450 uL of MACS buffer (PBS, 2% FBS, 1 mM EDTA)

and 50 µL of CD8a Microbeads (Miltenyi Biotec 130-117-044) were added per spleen and incubated at 4°C for 15 min. 20 mL of MACS buffer was then added to each tube and cells were pelleted as previously. The pellet was resuspended in 1–2 mL MACS buffer and transferred to a buffer-primed LS column in a magnetic holder with pre-separation filter. three column washes were carried out using 3 mL MACS buffer, and elution into a new tube (containing 5 mL of MACS buffer) using 5 mL of MACS buffer after removal from the magnetic holder, using the column plunger to force cells into the collection tube. 5 mL of MACS buffer was added to purified t-cells prior to counting. These cells were pelleted and resuspended in PBS at a concentration of $1*10^6$ cells/mL for injection.

## Tumor single-cell suspension

Tumor samples were dissected and minced with scissors, resuspended in 2 ml Digestion buffer composed of HBSS with 2 mg/ml Collagenase A (50-100-3278, Fischer Scientific) and 10 µg/ml DNase (D5025, Sigma-Aldrich), dissociated with a gentleMACS Dissociator (Miltenyi Biotec, 130-093-235) and incubated 30 min at 37°C. 10 ml PBS with 10% FCS (buffer A) was added and the suspension was passed through a 100 µm cell strainer (BD Biosciences). The resulting single-cell suspension was pelleted and resuspended in 1 ml ACK buffer (0.15 M $NH_4Cl$, 10 mM $KHCO_3$, 0.1 mM EDTA in distilled $H_2O$) and incubated on ice for 5 min. 10 ml of buffer A was added to stop the reaction. The suspension was passed through a 40-µm cell strainer (BD Biosciences), pelleted and resuspended in 500 µL of cold PBS. Single-cell suspensions were subsequently subject to cell phenotyping by flow cytometry as described below.

## Mouse CD8+ T-cell purification, activation, and expansion

Spleens were harvested, mashed over a 40-µm cell strainer (VWR, 10199–654), and CD8+ T-cells purified by positive magnetic bead selection (Miltenyi Biotec, 130-117-044) according to manufacturer's instructions. Purified CD8+ T-cells were counted and cell diameter measured using a Moxi Z mini automated counter (Orflo, MXZ001). The cells were characterized using Flow cytometry described below or plated at $1 \times 10^6$ (24-well plate) or $5 \times 10^5$ (48-well plate) and activated with Dynabeads Mouse T-Activator CD3/CD28 (Thermo Fisher, 11456D) in a 1:1 beat to T-cell ratio and cultured in 2 ml (24WP) or 1 ml (48WP) RPMI 1640 (Thermo Fisher, 21875) supplemented with 10% Fetal Bovine Serum, (Thermo Fisher, 10270–106), 50 µM 2-mercaptoethanol (Thermo Fisher, 21985023), 10 U/ml penicillin-streptomycin (Thermo Fisher, 15140122), and 10 U/ml recombinant human IL-2 (Sigma, 11011456001), and incubated at 37 °C for 3 days in a humidified $CO_2$ incubator.

## Cell proliferation analysis

The selected metabolites, citric acid (Sigma-Aldrich, 251275–100G), malic acid (Sigma-Aldrich, M1000-100G), succinic acid (Sigma-Aldrich, S3674-100G), fumaric acid (Sigma-Aldrich, 47910–100G), lactic acid (Sigma-Aldrich, L1750-10G), pyruvic acid (Sigma-Aldrich, 107360–25G), α-Ketoglutaric acid (Sigma-Aldrich, K1750), and oxaloacetic acid (Sigma-Aldrich, O4126) were prepared in RPMI 1640 (ThermoFisher, 21875), supplemented with 50 ml Fetal Bovine Serum (ThermoFisher, 10270106), 5.5 ml Penicillin-Streptomycin (ThermoFisher, 15140122) and 555 µl 2-ME (ThermoFisher, 21985023) and pH adjusted to 7.2–7.4, by adding sodium hydroxide (wvr, pHenomenal). The metabolites were added to a 96-well plate round-bottom in a serial dilution prior to T-cell activation, as described below.

Spleens were harvested from WT mice, mashed over a 40-µm cell strainer (VWR, 10199–654) and $CD8^+$ T cells were purified by magnetic bead selection (Milentyie Biotec, 130-117-044) according to manufacturer's instructions. Cells were counted (BioRad, TC20) and resuspended in complete media. $1 \times 10^5$ cells placed in a 96-well plate with added metabolites were activated in a 1:1 bead to cell ratio with Dynabeads Mouse T-Activator CD3/CD28 (ThermoFisher, 11456D) and 10 U/ml recombinant human IL-2 (Sigma, HIL2-RO). The treated cells were cultured in a humidified 5% $CO_2$ incubator for 3 days. To assess cell proliferation of the activated CD8 T-cells in the presence of these metabolites, a resazurin reduction assay (Sigma-Aldrich, R7017-1G) was carried out on day 3 of the experiment. In short, 100 µl of the treated cells were transferred to a black 96-well plate, (Corning, 3610) to which 20 µl of resazurin solution was added to a final concentration of 10 µg/ml and incubated for 3 hr. The relative fluorescence units (RFU), proportional to the amount of metabolically active cells, was determined by measuring the fluorescence intensity of the media in the black 96-well plate at

530/25 nm excitation wavelength and 590/35 nm emission wavelength with a plate reader (BioTek, Synergy HTX) at three time points: 2 hr, 2 hr 30 min, and 3 hr of incubation. Cell proliferation curves of treated cells was determined by subtracting the RFU values from the background control (media only) and the change of RFU values over time was calculated and compared to the negative control (media and cells).

## Cell phenotyping by flow cytometry

Single-cell suspensions were washed and stained with Fixable Near-IR Dead Cell Stain Kit (Thermo Fisher, L10119) followed by staining of extracellular antigens with fluorochrome labeled antibodies. The Fixation/Permeabilization Solution Kit (BD Biosciences, 554714) was used for exposing cytoplasmic antigens. Fluorochrome-labeled antibodies against mouse antigens was used in different combinations. CD44 (clone IM7), CD45.1 (clone A20), CD45.2 (clone 104), CD8 (clone 53–6.7), CTLA-4 (clone UC10-4F10-11), were purchased from BD Biosciences. CD62L (clone MEL-14), and ICOS (clone C398.4A) were purchased from BioLegend. Anti-mouse Granzyme B (clone GB12) was purchased from Thermo Fisher. Cell counting was performed with CountBright Absolute Counting Beads (Thermo Fisher, C36950). Samples were processed in a FACSCanto II flow cytometer (BD Biosciences). Data analysis was performed with FlowJo, version 8.8.7 (Tree Star).

## Cytotoxicity assay

Sodium L-Lactate (Sigma, L7022) and Sodium Chloride (Sigma, S5886) were prepared as 10x concentrated solutions in complete media. Compounds were added to purified OVA-specific OT-I CD8+ T-cells T-cells at the point of activation (day 0) minutes before addition of CD3/CD28 Dynabeads. Sodium chloride or plain media was used as control. OVA-expressing GFP-positive EL4 (EL4-GFP-OVA) cells were mixed with their respective parent cell line (EL4) in a 1:1 ratio in round-bottom 96-well plates. OVA-specific OT-I CD8+ T-cells were mixed with EL4 cells at effector to target ratios ranging from 20:1 to 1:50. Cytotoxicity was assessed by flow cytometry 24 hr later. The ratio of GFP-positive events (target) to GFP-negative events in each test co-culture was divided by the ratio from cultures without addition of OT-I cells to calculate specific cytotoxicity.

## Immunohistochemistry characterization of primary tumors

Mammary gland sections were deparaffinized with Tissue-Tek Tissue-Clear (Sakura, Japan) and rehydrated with graded ethanol. Antigen retrieval was performed either with Proteinase K (for F4/80) or boiling the sections in a high pH antigen retrieval buffer (Dako Target retrieval Solution, pH 9, Dako, Denmark) (for Podocalyxin, CD3, Granzyme B) or low pH antigen retrieval buffer (Dako Target retrieval Solution, pH 6, Dako, Denmark) (for PCNA). Endogenous peroxidase activity was blocked using 3% $H_2O_2$. Nonspecific protein interactions were blocked using 20% goat serum in PBS-T. Primary antibodies were used in mammary glands against CD3 (#ab5690, Abcam, UK) to assess lymphocytic infiltration, F4/80 (#MCA497, Abd Serotec, Germany) to assess macrophage density, podocalyxin (#AF1556, R and D Systems, Germany) to assess capillary density, PCNA (#M0879, Dako, Denmark) to identify proliferating cells, and Granzyme B (#ab4059, Abcam, UK) to identify effector lymphocytes. Subsequently the sections were incubated with a suitable biotin-conjugated secondary IgG antibody. All sections were then incubated with an Avidin-Biotin peroxidase Complex (ABC) kit (#PK6100, Vector, USA) and the peroxidase was stained with 3,3'-Diaminobenzidine(DAB) kit (#SK4100, Vector, USA). Nuclei were stained with hematoxylin and sections were mounted with Faramount Aqueous Mounting Medium, (Dako, Denmark). Specific stainings were quantified using Image J software (National Institute of Health, USA).

## Histological characterization

Mammary glands were stained with H and E with a standard protocol and histologically scored for tumor stage in accordance with (*Lin et al., 2003*), ranging from hyperplasia (score 1), adenoma/mammary intraepithelial neoplasia (adenoma/MIN) (score 2), early invasive carcinoma (score 3) to late invasive carcinoma (score 4). The highest staging value of each specimen was used.

## Organ harvest and metabolite extraction

For the metabolite analyses, blood and skeletal muscle (quadriceps) (Metabolite analysis 1) or blood together with five different tissue types (Metabolite analysis 2) were harvested from both exercising and control animals. Immediately after the exhaustion test, blood was collected from the mouse tail into a lithium heparin-covered microvette tube and kept at room temperature until further processing. Mice were subsequently euthanized by cervical dislocation prior to tissue collection. Biopsies of spleen, muscle, muscle-draining/non-draining lymph nodes and thymus were harvested and immediately frozen in liquid nitrogen. To assure that the right lymph nodes were collected, the muscle-draining lymph nodes had been identified in a previous experiment by injecting either the hind or the forelimb muscle of live animals with Patent Blue dye. The level of perfusion of local lymph nodes, under anesthesia, was then checked. The forelimb, muscle-draining lymph nodes showed to be the most easily accessible. Non-draining lymph nodes were harvested from the inguinal fat pad. Plasma was obtained by gradient centrifugation at 2000xg for 5 min at room temperature and stored at −80.

## GC/MS analysis

The initial metabolic profiling by GC-MS was performed at the Swedish Metabolomics Center in Umeå, Sweden. Information about reagents, solvents, standards, reference and tuning standards, and stable isotopes internal standards can be found as Supplementary information.

Sample Preparation: Extraction was performed as previously described (*Jiye et al., 2005*). For the plasma samples of extraction buffer (90/10 v/v methanol: water) including internal standards were added to the plasma samples. The volume of plasma varied from 25 µL to 100 µL and the volume of extraction solution was adjusted accordingly keeping the volume ratio of plasma to extraction solution constant (10/90 v/v plasma: extraction solution). The samples were shaken at 30 Hz for 2 min in a mixer mill and proteins were precipitated for 2 hr at −20°C. The samples were centrifuged at +4°C, 14,000 rpm, for 10 min. 200 µL (LC-MS) and 50 µL (GC-MS) of the supernatant was transferred to microvials and solvents were evaporated to dryness. For muscle tissue samples (all samples 20 mg + / - 2 mg with the individual weight of each sample noted), 1000 µl extraction buffer (90/10 v/v methanol: water) including internal standards was added to the tissue samples. To all tissue samples two tungsten beads were added and samples were shaken at 30 Hz for 3 min in a mixer mill. The samples were centrifuged at +4°C, 14,000 rpm, for 10 min. 50 µL (GC-MS) of the supernatant was transferred to micro vials and solvents were evaporated to dryness.

A small aliquot of the remaining supernatants were pooled and used to create quality control (QC) samples. The samples were analyzed in batches (different sample types) according to a randomized run order on GC-MS.

## UHPLC/MS analysis

Briefly, the second analysis was done on the Precision Metabolomics platform at Metabolon Inc (Morrisville, NC, US) samples were homogenized and subjected to methanol extraction then split into aliquots for analysis by ultrahigh performance liquid chromatography/mass spectrometry (UHPLC/MS) in the positive (two methods) and negative (two methods) mode. Metabolites were then identified by automated comparison of ion features to a reference library of chemical standards followed by visual inspection for quality control (as previously described, *Dehaven et al., 2010*). For statistical analyses and data display, any missing values are assumed to be below the limits of detection; these values were imputed with the compound minimum (minimum value imputation). Statistical tests were performed in ArrayStudio (Omicsoft) or 'R' to compare data between experimental groups; $p < 0.05$ is considered significant and $0.05 < p\ 0.10$ to be trending. An estimate of the false discovery rate (Q-value) is also calculated to take into account the multiple comparisons that normally occur in metabolomic-based studies, with $q < 0.05$ used as an indication of high confidence in a result.

## Isotopic labeling, metabolite extraction, and GC/MS analysis

Mouse CD8+ T cells, purified from spleen of vaccinated animals, directly after an exhaustion test, was washed with PBS, and metabolic activity quenched by freezing samples in dry ice and ethanol, and stored at −80 °C. Metabolites were extracted by addition of 600 µl ice-cold 1:1 (vol/vol)

methanol/water (containing 1 nmol scyllo-Inositol as internal standard) to the cell pellets, samples were transferred to a chilled microcentrifuge tube containing 300 µl chloroform and 600 µl methanol (1500 µl total, in 3:1:1 vol/vol methanol/water/chloroform). Samples were sonicated in a water bath for 8 min at 4°C, and centrifuged (13,000 rpm) for 10 min at 4°C. The supernatant containing the extract was transferred to a new tube for evaporation in a speed-vacuum centrifuge, resuspended in 3:3:1 (vol/vol/vol) methanol/water/chloroform (350 µl total) to phase separate polar metabolites (upper aqueous phase) from apolar metabolites (lower organic phase), and centrifuged. The aqueous phase was transferred to a new tube for evaporation in a speed-vacuum centrifuge, washed with 60 µl methanol, dried again, and derivatized by methoxiation (20 µl of 20 mg/ml methoxyamine in pyridine, RT overnight) and trimethylsilylation (20 µl of N,O-bis(trimetylsilyl)trifluoroacetamide + 1% trimethylchlorosilane) (Sigma, 33148) for $\geq$1 hr. GC/MS analysis was performed using an Agilent 7890B-5977A system equipped with a 30 m + 10 m $\times$ 0.25 mm DB-5MS + DG column (Agilent J and W) connected to an MS operating in electron-impact ionization (EI) mode. One microliter was injected in splitless mode at 270°C, with a helium carrier gas. The GC oven temperature was held at 70°C for 2 min and subsequently increased to 295°C at 12.5°C/min, then to 320°C at 25°C/min (held for 3 min). MassHunter Workstation (B.06.00 SP01, Agilent Technologies) was used for metabolite identification by comparison of retention times, mass spectra and responses of known amounts of authentic standards. Metabolite abundance and mass isotopologue distributions (MID) with correction of natural 13C abundance were determined by integrating the appropriate ion fragments using GAVIN.

## Statistics
Statistical analyses were performed with Prism 7 version 7.0 (GraphPad). Statistical tests and number of replicates are stated in figure legends. Principal component analysis (PCA) was carried out with normalized metabolite concentrations using the *prcomp* function in R version 3.6.1. Fold changes and hierarchical clustering of metabolites was performed using the *hclust* function in R version 3.6.1.

## Study approval
All animal experiments were approved by the regional animal ethics Committee of Northern Stockholm, Sweden (Dnr: N78/15 and N101/16). In the human study, all subjects were informed about the study outline, gave written informed consent prior to inclusion and was familiarized with the experimental procedures. All procedures were performed in accordance with the Helsinki declaration and ethical standards of the institutional and national research committee Dnr: 2016/590–31.

## Acknowledgements
Carolin Lindholm and Andrea Berk (Karolinska Institutet) for expertise in immunohistochemistry, Prof. Kristian Pietras (Lunds Universitet) for sharing the MMTV-PyMT mouse strain on the FVB background, Dr. Carina Strell (Karolinska Institutet) for sharing the I3TC cell line, and Prof. Ananda Goldrath and Prof. Christer Höög for discussions. Metabolomics experiments were performed at the Swedish Metabolomics Center (Umeå, Sweden) and on the Precision Metabolomics platform at Metabolon Inc (Morrisville, NC, US). [U-$^{13}$C$_6$]glucose metabolomics experiments were performed in the Metabolomics STP at The Francis Crick Institute (London, UK), which receives its core funding from Cancer Research UK (FC001999), the UK Medical Research Council (FC001999), and the Wellcome Trust (FC001999). The work was funded by the Knut and Alice Wallenberg Scholar Award, the Swedish Medical Research Council (Vetenskapsrådet), the Swedish Cancer Fund (Cancerfonden), the Swedish Children's Cancer Fund (Barncancerfonden), the Swedish Society of Medicine (Svenska läkaresällskapet) and a Principal Research Fellowship (214283/Z/18/Z) to RSJ from the Wellcome Trust. We thank James MacRae and James Ellis for help with the GC-MS analysis.

## Additional information

### Funding

| Funder | Grant reference number | Author |
|---|---|---|
| Knut och Alice Wallenbergs Stiftelse | Wallenberg Scholar | Randall Johnson |
| Vetenskapsrådet | | Randall Johnson |
| Cancerfonden | | Randall Johnson |
| Barncancerfonden | | Randall Johnson |
| Svenska Läkaresällskapet | | Helene Rundqvist |
| Cancer Research UK | | Paulo A Gameiro |
| Medical Research Council | | Jernej Ule |
| Wellcome Trust | Principal Research Fellowship 214283/Z/18/Z | Randall Johnson |

The funders had no role in study design, data collection and interpretation, or the decision to submit the work for publication.

### Author contributions

Helene Rundqvist, Conceptualization, Formal analysis, Investigation, Methodology, Writing - original draft, Writing - review and editing; Pedro Veliça, Conceptualization, Investigation, Methodology, Writing - original draft, Writing - review and editing; Laura Barbieri, Paulo A Gameiro, Milos Gojkovic, David Wulliman, Emil Ahlstedt, Jernej Ule, Investigation; David Bargiela, Data curation, Investigation, Writing - original draft; Sara Mijwel, Stefan Markus Reitzner, Investigation, Writing - original draft; Arne Östman, Supervision, Funding acquisition, Investigation, Writing - original draft, Project administration; Randall S Johnson, Conceptualization, Data curation, Formal analysis, Supervision, Funding acquisition, Investigation, Methodology, Writing - original draft, Project administration, Writing - review and editing

### Author ORCIDs

Helene Rundqvist (iD) https://orcid.org/0000-0002-5617-9076
Pedro Veliça (iD) http://orcid.org/0000-0002-0557-8544
Laura Barbieri (iD) https://orcid.org/0000-0001-9932-2664
Sara Mijwel (iD) http://orcid.org/0000-0002-6718-5797
Stefan Markus Reitzner (iD) http://orcid.org/0000-0003-0151-2780
David Wulliman (iD) http://orcid.org/0000-0003-1632-0394
Jernej Ule (iD) http://orcid.org/0000-0002-2452-4277
Randall S Johnson (iD) https://orcid.org/0000-0002-4084-6639

### Ethics

Human subjects: Informed consent was obtained. In the human study, all subjects were informed about the study outline, gave written informed consent prior to inclusion and was familiarized with the experimental procedures. All procedures were performed in accordance with the Helsinki declaration and ethical standards of the institutional and national research committee- Ethical permit Dnr: 2016/590-31.
Animal experimentation: The study was performed as approved by the regional Animal Ethics Board according to the laws of Sweden (Ethics approval: DNR78/15).

### Decision letter and Author response

Decision letter https://doi.org/10.7554/eLife.59996.sa1
Author response https://doi.org/10.7554/eLife.59996.sa2

## Additional files

### Supplementary files
• Transparent reporting form

### Data availability
All data generated are included in the manuscript and supporting files.

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
