## [Decision Letter]

**Acceptance summary:**

This study examined exercise-induced protection from tumors and showed that on the FVB background and with the right tumor cell line exercise protects in a CD8 T cell-dependent manner. Using metabolomics, the study focused on lactate as a potential mediator to increase CD8 T cell function and anti-tumor activities. This is supported by studies in which mice with tumors are injected with lactate and partially protected as tumor growth slows.

**Decision letter after peer review:**

Thank you for submitting your article "Cytotoxic T-cells mediate an exercise-induced reduction in tumor growth" for consideration by *eLife*. Your article has been reviewed by three peer reviewers, one of whom is a member of our Board of Reviewing Editors, and the evaluation has been overseen by Tadatsugu Taniguchi as the Senior Editor. The following individual involved in review of your submission has agreed to reveal their identity: Randy Hampton (Reviewer #2).

The reviewers have discussed the reviews with one another and the Reviewing Editor has drafted this decision to help you prepare a revised submission.

Essential revisions:

1) In Figure 3, the authors test the effects of a number of exercise-induced metabolites on CD8+ T cells in vitro. One major concern of this experimental approach is that the authors should show or state the physiological levels of those metabolites in vivo (e.g. serum levels), and then use proper ranges of concentrations for the in vitro experiments.

In Figure 3B, most metabolites (malate, succinate, fumarate, lactate and aKG) exert their functions at 1.56-3.13 mM concentrations. The authors should confirm whether the concentrations of those metabolites are within the ranges of their physiological levels. In addition, in Figure 3E, the authors used 40 mM lactate for the CD8+ T cell cytotoxicity assay, seemingly high relative to the known physiological concentration of lactate.

2) In addition to tumor tissues, does exercise affect CD8+ T cell populations in other lymphoid tissues? According to Figure 1E, the authors state that "Here, only CD8+ T cells showed a significant increase in frequency across these three tissues in the running animals". However, statistically significant differences were observed only in "Tumor" group. The authors should clearly show whether the differences in "Spleen" and "LN" groups are statistically significant, which is important for clarifying whether the effects of exercise on CD8+ T cell population are global or specific to tumor tissues.

3) Other proposed mechanisms of exercise-induced regulation of tumors, such as modulation of IL6, are not really addressed. It would be helpful for the authors to measure this cytokine in response to their exercise regimens.

4) In Figure 3B-D, the lactate and sodium L-lactate display different effects on CD8+ T cells. Since lactate and sodium L-lactate are similar metabolites after pH neutralization, the authors should explain why they show different functions. In addition, what is the physiological form of lactate in vivo, lactate or sodium L-lactate? I think the authors should use the physiological one to do the mechanistic and functional study.

5) For Figure 3C and D, the authors should clarify the underlying mechanisms by which sodium L-lactate increases the expressions of iCOS and Granzyme B in CD8+ T cells. Moreover, do the exercise-induced metabolites affect the expressions of other co-stimulatory molecules in CD8+ T cells such as CD28, PD-1 or CTLA-4?

6) In Figure 6, the authors show the anti-tumor function of sodium L-lactate in vivo and demonstrate that both CD4+ and CD8+ T cell populations are increased after daily sodium L-lactate treatment. The authors may further characterize the functions of sodium L-lactate in vivo. For example, does sodium L-lactate affect the expression of iCOS and Granzyme B in CD8+ T cells in vivo?

---

## [Author Response]

Essential revisions:1) In Figure 3, the authors test the effects of a number of exercise-induced metabolites on CD8+ T cells in vitro. One major concern of this experimental approach is that the authors should show or state the physiological levels of those metabolites in vivo (e.g. serum levels), and then use proper ranges of concentrations for the in vitro experiments.In Figure 3B, most metabolites (malate, succinate, fumarate, lactate and aKG) exert their functions at 1.56-3.13 mM concentrations. The authors should confirm whether the concentrations of those metabolites are within the ranges of their physiological levels. In addition, in Figure 3E, the authors used 40 mM lactate for the CD8+ T cell cytotoxicity assay, seemingly high relative to the known physiological concentration of lactate.

In response to the reviewers request we have included a table of resting, adult, human plasma levels of relevant metabolites in Figure 3—figure supplement 1A.

Data from the current manuscript unfortunately does not allow comparison of absolute levels between organs, and while data from the plasma represents primarily extracellular levels, metabolite concentrations from targeted organs represent a mix of intra- and extra-cellular levels. The comparison between exercised and non-exercised samples however, control for this, and allow us to detect exercise-induced differences within each organ. Box plots from the current manuscript are provided below, showing increases in plasma metabolites, and supporting a substantial local release. In addition, post-exercise increases of lactate (16-fold), pyruvate (8-fold), succinate (4-fold) and citrate (2-fold), relative to resting levels, has been described in the recent systematic review by Schranner, et al.^1^.

**Author response image 1. sa2fig1:** 

In the current manuscript, one of our primary findings is that systemic (plasma) levels of TCA metabolites increase significantly in response to acute exercise. Serum increases in the ranges of 28 fold suggest that at peripheral sites substantial amounts have been released. We agree with the reviewers that the in vitro responses of CD8+ T-cells shown in Figure 3 are in part at levels higher than commonly detected in plasma; however, we propose that at certain times and in some situations, e.g., when exposed to lymphatic fluid from muscle drainage post-exercise, the local extracellular concentration of TCA metabolites extends to these higher ranges.The manuscript now reads:

“To test this, we activated CD8+ T cells ex vivo for 3 days in the presence of increasing doses (25µM-50mM) of pH-neutralized lactic acid, pyruvic acid, citric acid, malic acid, succinic acid, fumaric acid, a-ketoglutaric acid, or sodium L-lactate. Reference human resting serum levels are provided in Figure 3—figure supplement 1A.” and

*“*Our findings that circulating levels of TCA metabolites increase by 2-8 fold after exercise, suggests that at peripheral sites, substantial amounts of these metabolites have been released into the lymphatic fluid draining the muscle tissue. […] Thus, to mimic the activation of T cells in a lymphoid organ exposed to the physiologically buffered metabolites generated by exercising muscle, we activated CD8+ T cells in the presence of TCA metabolites and lactate in media adjusted to a physiological pH.”

2) In addition to tumor tissues, does exercise affect CD8+ T cell populations in other lymphoid tissues? According to Figure 1E, the authors state that "Here, only CD8^+^ T cells showed a significant increase in frequency across these three tissues in the running animals". However, statistically significant differences were observed only in "Tumor" group. The authors should clearly show whether the differences in "Spleen" and "LN" groups are statistically significant, which is important for clarifying whether the effects of exercise on CD8^+^ T cell population are global or specific to tumor tissues.

The statistical analysis for Figure 1E is made as a column factor two-way ANOVA that takes all tissues into account (this has now been clarified in the figure legend).

3) Other proposed mechanisms of exercise-induced regulation of tumors, such as modulation of IL6, are not really addressed. It would be helpful for the authors to measure this cytokine in response to their exercise regimens.

There is substantial literature about the release of IL-6 from skeletal muscle, primarily in humans and in response to longer duration exercise. The Cell Metabolism paper by the late Professor Hojman’s group, from 2016, reports higher levels of IL-6 in exercising animals. We have carried out a cytokine array analysis of the PyMT model, data included in Author response image 2 and 3, and the trend is an increased level of IL-6 in tumor-bearing animals, and a reduction of steady state IL-6 levels in exercising animals, both non-tumor-bearing and tumor-bearing.

To investigate the acute effects of treadmill exercise on systemic IL-6 levels, we conducted a treadmill exhaustive exercise protocol, and measured lactate from the tail vein and IL-6 in heparin plasma from heart puncture, both directly after exercise. The control group in this case consisted of non-exercising animals. In this experiment we see no difference in systemic IL-6 levels, although the variability is somewhat reduced in the exercising animals; this is similar to the findings of Hojman et al. Although we cannot rule out IL-6 (or any other cytokine-mediated effect) as a contributor to mobilization or recruitment of immune cells in response to exercise, we find this beyond the scope of the current manuscript. As expected, lactate levels increase in response to the acute exercise, and the lactate data is now included in Figure 2—figure supplement 1G

**Author response image 2. sa2fig2:** Cytokine array (RayBiotech, Norcross, GA, USA), describing the cytokine profile in serum from WT and tumor bearing (PyMT) running and non-running mice. Chemiluminiscens detection was done on the ImageQuant LAS 4000 (GE Healthcare, Buckinghamshire, UK).

**Author response image 3. sa2fig3:** Left panel, mouse IL-6 plasma (RnD ELISA, heart puncture) levels after an acute treadmill exercise and in control animals. Right panel, blood (tail vein) lactate in the same cohort (Accutrend device).

4) In Figure 3B-D, the lactate and sodium L-lactate display different effects on CD8+ T cells. Since lactate and sodium L-lactate are similar metabolites after pH neutralization, the authors should explain why they show different functions. In addition, what is the physiological form of lactate in vivo, lactate or sodium L-lactate? I think the authors should use the physiological one to do the mechanistic and functional study.

In the physiological setting, the excess H+ seen in conjunction with lactic acid production will be buffered by a range of different in vivo buffering systems, including phosphate and bicarbonate. In an attempt to model an increased lactate availability without significantly altering pH, we used the two different strategies shown in Figure 3 (Lactic acid buffered to pH 7.35 with NaOH, and Sodium Lactate). The difference in T-cell response is indeed puzzling but may illuminate an aspect of how lactate exerts its effect on cytotoxic T-cells, a subject currently under intense investigation in the lab.

5) For Figure 3C and D, the authors should clarify the underlying mechanisms by which sodium L-lactate increases the expressions of iCOS and Granzyme B in CD8+ T cells. Moreover, do the exercise-induced metabolites affect the expressions of other co-stimulatory molecules in CD8+ T cells such as CD28, PD-1 or CTLA-4?

The underlying mechanisms explaining the functional changes found in T-cells in response to metabolite exposure is the primary focus of multiple research consortia. For example, in a recent Nature Metabolism publication from Erika Pearce’s group, glucose deprivation alters the function of CD8+ T-cells and leads to an improved anti-tumor function^2^; there, the mechanism proposed is that re-exposure to glucose enhances the PPP, and thereby allows for increased expansion. We are currently exploring a number of models to explain the overall role of lactate in different microenvironmental settings, with a view to understanding how it feeds into T cell metabolism and differentially influences it under differing pH and oxygenation states. A more extensive explanation of this is being explored intensively by our laboratory at the moment and will certainly be the subject of future work from our lab and others.

Information about CTLA-4 expression in response to increasing doses of central carbon metabolites is now included in Figure 3—figure supplement 1G.

The manuscript has been changed accordingly:

”Furthermore, sodium L-lactate induced a dose-dependent increase in inducible T-cell costimulator (iCOS) and Granzyme B (GzmB) expression at day 3 of culture (Figure 3C-D, Figure 3—figure supplement 1E-F). We found no significant effect of metabolite exposure on T-cell CTLA-4 expression (Figure 3—figure supplement 1G)”.

6) In Figure 6, the authors show the anti-tumor function of sodium L-lactate in vivo and demonstrate that both CD4+ and CD8+ T cell populations are increased after daily sodium L-lactate treatment. The authors may further characterize the functions of sodium L-lactate in vivo. For example, does sodium L-lactate affect the expression of iCOS and Granzyme B in CD8+ T cells in vivo?

We have now extended the information on the tumor infiltrating CD8+ population by analyzing GzmB expressing cell fractions and MFI of GzmB, PD1 and CTLA4. In these experiments we found no difference in effector molecule expression after daily lactate injections, despite increasing number of CD8+ T-cells. Data is provided below and included in the manuscript as Figure 6—figure supplement 1C-E.

The text has been changed accordingly:

“As shown in Figure 6B, there were significant changes in tumor infiltrating immune populations in animals treated with 2 g/kg of lactate, namely, increases in the frequency of total (CD3+) T cells, including both CD4+ and CD8+ T cells. The frequency of tumor-infiltrating NK cells was reduced. Despite finding increased numbers of tumor infiltrating CD8+ T cells (Figure 6B and Figure 6—figure supplement 1C), daily lactate administration did not change the percentage of GzmB+ cells or the GzmB, PD1 or CTLA intensity on CD8+ infiltrating cells (Figure 6—figure supplement 1D-E)”.

References:

1) Schranner, D., Kastenmuller, G., Schonfelder, M., Romisch-Margl, W. and Wackerhage, H. Metabolite Concentration Changes in Humans After a Bout of Exercise: a Systematic Review of Exercise Metabolomics Studies. Sports Med Open 6, 11, doi:10.1186/s40798-020-0238-4 (2020).

2) Klein Geltink, R. I. et al. Metabolic conditioning of CD8(+) effector T cells for adoptive cell therapy. Nat Metab 2, 703-716, doi:10.1038/s42255-020-0256-z (2020).